# From Prior to Pro:
# Efficient Skill Mastery via Distribution Contractive RL Finetuning

**Zhanyi Sun**[1]  **Shuran Song**[1]

## Abstract

We introduce Distribution Contractive Reinforcement Learning (DICE-RL), a framework that uses reinforcement learning (RL) as a "distribution contraction" operator to refine pretrained generative robot policies. DICE-RL turns a pretrained behavior prior into a high-performing "pro" policy by amplifying high-success behaviors from online feedback. We pretrain a diffusion- or flow-based policy for broad behavioral coverage, then finetune it with a stable, sample-efficient residual off-policy RL framework that combines selective behavior regularization with value-guided action selection. Extensive experiments and analyses show that DICE-RL reliably improves performance with strong stability and sample efficiency. It enables mastery of complex long-horizon manipulation skills directly from high-dimensional pixel inputs, both in simulation and on a real robot. Project website: dice.rl.2026.

## 1. Introduction

What role should reinforcement learning (RL) play in post-training robot policies? In this work, we focus on *sparse-reward, long-horizon* manipulation settings where online interaction is expensive and unconstrained exploration is infeasible. Under these constraints, we argue that RL is most effective and practical when used as a *"distribution contractor"* on top of a pretrained generative behavior cloning (BC) policy: starting from a policy that already produces physically plausible behaviors, RL can *reweight* its action distribution using online feedback, increasing the probability of high-success behaviors while suppressing failure-prone ones. This perspective is inspired by post-training in large language models, where reinforcement learning with verifiable rewards (RLVR) sharpens a pretrained model by am-

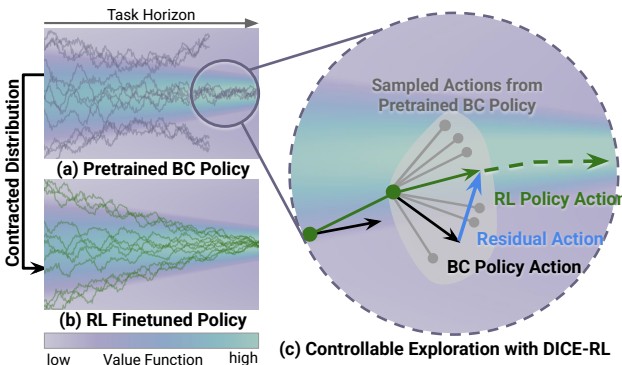

*Figure 1.* **Distribution Contractive RL (DICE-RL)** refines (a) a pretrained generative BC policy (i.e., the behavior prior) into (b) a "pro" policy by contracting the action distribution around successful action modes. DICE-RL leverages the generative behavior cloning policy to achieve (c) controllable exploration for sample-efficient and stable reinforcement learning.

plifying responses that satisfy task-specific checks (Huang et al., 2024; Zhao et al., 2025; Yue et al., 2025).

Translating this idea to robotics is nontrivial. Robotics involves *continuous* action space and *costly* verification that requires physical execution, while rewards are delayed and horizons are long. Under tight online interaction budgets, the central challenge becomes **efficient, controllable exploration**: exploration must be rich enough to correct systematic BC failures, yet constrained enough to avoid drifting far from the pretrained policy. This motivates two principles for effective contraction: (1) the pretrained policy should already cover viable solutions in action space (even if imprecise or stochastic), and (2) RL post-training should improve performance by contracting behavior *within* the pretrained policy's support. Guided by these principles, we propose **Distribution Contractive RL Finetuning (DICE-RL)**, which addresses the following questions:

**How to provide a useful behavior prior?** Effective finetuning relies on a pretrained policy that provides *rich* action proposals that remain physically plausible. We therefore pretrain a diffusion-based BC policy on offline demonstrations and use early stopping to avoid overfitting and preserve diversity. Such generative policies can represent complex action distributions and naturally support stochastic inference, yielding a behavior prior that generalizes across states

[1]Stanford University. Correspondence to: Zhanyi Sun <zhanyis@stanford.edu>.

*Proceedings of the 43rd International Conference on Machine Learning*, Seoul, South Korea. PMLR 306, 2026. Copyright 2026 by the author(s).

and provides structured exploration during RL finetuning.

**How to achieve controllable exploration under limited interaction budgets?** Given a stochastic generative behavior prior, the remaining challenge is to correct its systematic failures without destabilizing learning or drifting arbitrarily in continuous action spaces, all under tight interaction budgets. We address this by parameterizing the finetuned policy as a lightweight *residual* on top of the frozen BC prior, so that RL updates act as local action corrections around the prior's proposals; this preserves the prior's expressiveness and reduces the effective search space.

To keep exploration controllable, we introduce *selective behavior regularization*: we apply a BC-style penalty that pulls the residual policy toward the pretrained prior in states where the prior already achieves high value, and relax this penalty only at states where higher-return behaviors have been observed during online finetuning. Finally, to mitigate occasional low-value samples from a stochastic policy during interaction, we apply *value-guided action selection* by scoring a set of candidate action samples and executing the highest-valued one. Together, these mechanisms yield stable, sample-efficient contraction of the policy distribution toward successful behaviors while keeping exploration largely within the prior's support.

In summary, this paper makes three contributions:

- **A practical RL finetuning framework for generative BC policies.** We propose DICE-RL, a stable and sample-efficient off-policy RL finetuning framework for diffusion-based BC policies tailored to sparse-reward, long-horizon manipulation.
- **Strong empirical results in simulation and on a real robot.** DICE-RL achieves strong performance on challenging long-horizon manipulation tasks from raw visual observations, both in simulation and on a real robot.
- **Understanding and guidance for policy post-training.** We analyze the effects of RL post-training on generative BC policies (e.g., distribution sharpening/contraction) and conduct systematic studies of key pre- and post-training strategies (data properties, learning formulation, and training procedures) to provide actionable guidance.

## 2. Related Work

A growing body of work applies reinforcement learning (RL) to pretrained behavior cloning (BC) policies. In this context, we review different strategies proposed for BC pretraining and RL post-training.

**Offline BC Pretraining.** Pretraining is typically framed as supervised behavior cloning (BC) on large-scale offline datasets and has shown broad success in robotics (Argall et al., 2009; Ross et al., 2011; Bojarski et al., 2016; Rahmatizadeh et al., 2018; Shafiullah et al., 2022; Brohan et al.,

2022; Kim et al., 2024). However, naive BC can suffer from covariate shift and compounding error (Ross et al., 2011; Laskey et al., 2017; Ke et al., 2023; Sun & Song, 2025; Simchowitz et al., 2025), motivating RL post-training as a refinement step. Recently, diffusion- and flow-based policies (Chi et al., 2024; Black et al., 2024) have become popular BC backbones: their expressiveness supports modeling complex action distributions, but their iterative computation complicates direct integration with standard RL formulations. Moreover, while BC has been extensively studied for imitation performance, its role as a *pretraining mechanism* for downstream RL finetuning is comparatively less studied (Chen et al., 2025; Wagenmaker et al., 2025a). In this work, we adopt diffusion- and flow-based policies for offline pretraining and systematically study how pretraining choices (model, data, and training procedure) affect downstream finetunability, alongside a finetuning method designed to operate effectively with iterative generative policies.

**RL Finetuning of Generative BC Policies.** A wide range of methods improve pretrained policies using offline data (Chen et al., 2022; Hansen-Estruch et al., 2023; Wang et al., 2022; Chen et al., 2023; Kang et al., 2023) or online interaction (Ball et al., 2023; Zhang et al., 2023; Hu et al., 2023; Xu et al., 2022; Haldar et al., 2023; Mendonca et al., 2024; Yang et al., 2024; Gupta et al., 2021; Sharma et al., 2023; Zhu et al., 2020; Luo et al., 2024). Here, we focus on approaches that explicitly use diffusion- and flow-based models as the pretrained BC policy. One line of work directly **finetunes** the generative model parameters using reinforcement learning. Several methods adapt policy gradient algorithms to diffusion or flow-based policies, either by modeling the denoising chain as a Markov Decision Process with tractable per-step likelihoods (Ren et al., 2024; Ding et al., 2025; Lei et al., 2025) or by using the flow matching loss as a surrogate for the policy ratio (McAllister et al., 2025). Others employ reward-weighted regression or group-relative optimization to bias the generative training objective toward high-reward samples (Pfrommer et al., 2025). A second line of work uses **distillation**. These methods either learn a one-step actor with TD-style objectives (Park et al., 2025; Li et al., 2025) or use value-guided action selection to generate improved targets that are distilled back into the policy (Dong et al., 2025). By avoiding backpropagation through denoising, distillation simplifies post-training and can improve stability. A third line of work focuses on **steering** or **correction** of a fixed pretrained policy. Steering methods guide sampling at test time (Frans et al., 2025; Wagenmaker et al., 2025b) or learn noise-selection mechanisms (Wagenmaker et al., 2025b), while correction methods learn lightweight residual modules that locally adjust the base policy's outputs (Yuan et al., 2024; Ankile et al., 2025b;a). Keeping the base policy fixed improves stability,

but steering remains bounded by the base policy's failure modes, whereas residual correction can extend behavior beyond the base policy when needed.

***Our approach strategically integrates and extends multiple ideas from prior work in a unique and practical way*** – it inherits the stability of distillation-style approaches (via BC regularization), the efficiency of the steering-style approaches (via value-guided action selection), and the flexibility of correction-style approaches via residual action learning. As a result, DICE-RL can explicitly target a sharpened and more reliable action distribution for long-horizon manipulation, while allowing controlled deviations from the pretrained BC policy to correct systematic errors.

## 3. Preliminaries

We consider an MDP $\mathcal{M} = (\mathcal{S}, \mathcal{A}, \mathcal{P}, p_0, r, \gamma)$ with transition kernel $\mathcal{P}(\cdot \mid s, a)$, initial-state distribution $p_0$, reward function $r : \mathcal{S} \times \mathcal{A} \to \mathbb{R}$, and discount factor $\gamma \in [0, 1]$. A policy $\pi$ aims to maximize the expected discounted return $\mathbb{E}_\pi \left[ \sum_{t=0}^T \gamma^t r(s_t, a_t) \right]$. We study sparse-reward manipulation tasks, where success is reflected only at the end of an episode. We assume access to an offline demonstration dataset $D_{\text{demo}}$ and maintain an online replay buffer $D_{\text{online}}$ for experience collected during finetuning.

**Diffusion- and flow-based BC policies.** Behavioral cloning can be viewed as learning a state-conditioned generative model of actions from demonstrations $(s, a) \sim D_{\text{demo}}$. Both diffusion policies (e.g., trained with DDPM loss and sampled with DDIM) and flow-matching policies instantiate this by transforming *latent noise* $z \sim \mathcal{N}(0, I)$ into an action conditioned on $s$. We denote the resulting deterministic sampling map by

$$\pi_{\text{pre}}(s, z) \triangleq \psi_{\theta_{\text{pre}}}(s, z), \qquad z \sim \mathcal{N}(0, I), \qquad (1)$$

where $\psi_\theta$ is the solution map of the generative dynamics: it can be obtained either by (i) iterating a denoising recursion $x_{t-1} = \epsilon_\theta(t, s, x_t)$ from $x_T = z$ to $x_0$ in diffusion policies, or (ii) integrating a conditional velocity field $\dot{x} = v_\theta(t, s, x)$ from $x(0) = z$ to $x(1)$ in flow matching policies. Thus, $\pi_{\text{pre}}(s, z)$ is deterministic given $(s, z)$ and induces stochasticity only through sampling $z$. The majority of our experiments use a flow-matching backbone for the pretrained policy, and we adopt flow-matching terminology throughout for simplicity; however, DICE-RL is equally applicable when the pretrained policy is implemented as a diffusion policy.

## 4. DICE-RL

We build on a pretrained flow matching BC policy $\pi_{\text{pre}}(s, z)$ and *never* update its parameters. Instead of finetuning the

generative model itself (which would require differentiating through the ODE solver and can be costly and unstable), we treat $\pi_{\text{pre}}$ as a fixed stochastic proposal distribution: sampling $z \sim \mathcal{N}(0, I)$ yields structured exploration within the support of demonstrations. We represent the RL policy as a lightweight residual $s_\theta(s, z)$ applied to an $h$-step *action chunk*,

$$a_{t:t+h-1} = \pi_{\text{pre}}(s_t, z) + s_\theta(s_t, z), \qquad z \sim \mathcal{N}(0, I), \tag{2}$$

and learn an ensemble critic $\{Q_{\phi_i}\}_{i=1}^{N_Q}$ over action chunks. Because $\pi_{\text{pre}}(s_t, z)$ is deterministic for a fixed latent noise $z$, conditioning the residual policy $s_\theta(s_t, z)$ on the same $z$ makes the correction explicitly aware of the particular base action chunk proposed by $\pi_{\text{pre}}$.

The residual parameterization has two practical benefits. First, it preserves the pretrained flow policy's expressive, stochastic action generation, and learns only a lightweight residual correction policy $s_\theta(s_t, z)$ on top. This avoids iterative denoising during RL optimization and allows straightforward reparameterized policy-gradient updates through the residual. Second, it provides an explicit mechanism for controllable exploration within the demonstrations' support: we regularize the residual magnitude so that, by default, the policy stays close to $\pi_{\text{pre}}$ and only makes small value-improving edits. Concretely, we train the residual actor with a TD3+BC-style objective (Fujimoto & Gu, 2021),

$$\min_\theta \mathbb{E}_{\substack{s \sim \mathcal{D} \\ z \sim \mathcal{N}(0, I)}} \left[ -Q_\phi(s, a) + \beta \|s_\theta(s, z)\|_2^2 \right], \quad (3)$$

where $a = \pi_{\text{pre}}(s, z) + s_\theta(s, z) \in \mathcal{A}^h$ denotes an action chunk (Eq. (2)).

The first term maximizes value under the critic and the second term is a BC-style regularization loss that encourages exploration within the pretrained policy's support. We later introduce a filter that selectively disables this regularizer when the action from the finetuned RL policy is reliably value-improving. During online RL finetuning, we freeze the observation encoder learned during BC pretraining, using it to map high-dimensional observations into a compact latent feature space for RL learning. Algo. 1 summarizes the full training procedure; below we highlight the key design choices that enable sample-efficient and stable finetuning.

**Action chunking.** Action chunking is now standard in offline behavior cloning (Zhao et al., 2023; Chi et al., 2024; Simchowitz et al., 2025; Zhang et al., 2025) and has recently been shown to improve reinforcement learning as well (Huang et al., 2025; Li et al., 2025). We adopt this in our setting by applying residual finetuning at the chunk level (Eq. (2)) and train a chunk critic $Q_\phi(s_t, a_{t:t+h-1})$ with $h$-step bootstrapping (Eq. (4)). Action chunking improves temporal consistency and reduces the effective decision frequency, which is particularly helpful for long-horizon

manipulation where sparse rewards make per-step credit assignment noisy and inefficient.

**Adaptive RLPD mixing.** DICE-RL finetunes from a mixture of offline-to-online data, sampling mini-batches from $D_{\text{demo}}$ and $D_{\text{online}}$ with an RLPD-style ratio $r_{\text{offline}}(t) \in [0, 1]$ at environment step $t$. Each update draws data from $r_{\text{offline}}(t) D_{\text{demo}} + (1 - r_{\text{offline}}(t)) D_{\text{online}}$. Instead of keeping a fixed ratio as done in the original RLPD paper (Ball et al., 2023), we employ a linear decay schedule: $r_{\text{offline}}(t)$ decreases from $r_{\text{offline}}^{\text{start}}$ to $r_{\text{offline}}^{\text{end}}$ over the first $T_{\text{ratio}}$ steps and stays at $r_{\text{offline}}^{\text{end}}$ afterwards. This schedule anchors learning to demonstrations early for stability, while gradually shifting weight to online experience as the residual improves. $T_{\text{ratio}}$ is set to span the initial warm-start period, when online data are sparse and the residual is still rapidly changing; beyond this point, the policy updates are primarily driven by online experience. While DICE-RL uses offline data by default, the ablation in Appendix B indicates that using less (or even no) offline data has only a minor effect on final finetuning performance, suggesting the RLPD schedule mainly improves early stability rather than being strictly necessary.

**Multi-sample Expectation Training.** The pretrained flow matching policy induces a structured action *distribution* at each state via its latent $z$. Rather than collapsing this stochasticity into a single sampled action during training, we optimize objectives that are explicitly averaged over latent samples. This has two benefits: (i) it lets the residual improve the entire latent-induced action distribution of $\pi_{\text{pre}}$ instead of overfitting to one draw, and (ii) it provides a low-variance, sample-efficient training signal by reusing $K$ candidates per visited state. Concretely, at a state $s$ from a minibatch we draw $\{z_k\}_{k=1}^K$ and form $K$ chunk candidates $a^{(k)} = \pi_{\text{pre}}(s, z_k) + s_\theta(s, z_k) \in \mathcal{A}^h$. We train the critic with an $h$-step TD target that bootstraps from the *average* value over $K$ next-state candidates:

$$y = \sum_{j=0}^{h-1} r_{t+j} + \gamma \cdot \frac{1}{K} \sum_{k=1}^K Q_{\phi'}\big(s_{t+h}, a_{t+h:t+2h-1}^{\prime(k)}\big), \quad (4)$$

where $a_{t+h:t+2h-1}^{\prime(k)} = \pi_{\text{pre}}(s_{t+h}, z_k') + s_\theta(s_{t+h}, z_k')$ and $z_k' \sim \mathcal{N}(0, I)$. For the actor, we maximize the critic value *averaged* over the $K$ candidates at the current state:

$$\mathcal{L}_{\text{RL}}(\theta) = -\frac{1}{K} \sum_{k=1}^K Q_\phi\big(s_t, a_{t:t+h-1}^{(k)}\big). \quad (5)$$

During online interaction, we perform best-of-$N$ action selection: we sample $\{z_k\}_{k=1}^K$, form candidate action chunks $a_{t:t+h-1}^{(k)} = \pi_{\text{pre}}(s_t, z_k) + s_\theta(s_t, z_k)$, and execute the highest-valued candidate $k^\star = \arg\max_k Q_\phi(s_t, a_{t:t+h-1}^{(k)})$.

**BC loss filter.** The residual penalty $\|s_\theta(s, z)\|_2^2$ keeps fine-tuning conservative, but applying it uniformly can also suppress necessary deviations: when an edited action is *truly* better than the base sample, we would like to stop pulling it back toward 0 and allow RL to retain the improvement (Haldar et al., 2023). However, a learned critic can be *optimistically biased*, especially early in training. If we disable regularization whenever the critic predicts a gain, the actor can exploit spurious Q overestimation and drift away from the pretrained support. To prevent this, we use a simple heuristic that relaxes the BC penalty only when the critic predicts the residual action improves upon the base action *and* this predicted value does not exceed a Monte-Carlo return estimate (up to a small negative margin $\epsilon$). For $(s, z)$, let $a^{\text{pre}} = \pi_{\text{pre}}(s, z)$ and $a^{\text{cur}} = \pi_{\text{pre}}(s, z) + s_\theta(s, z)$, and let $\widehat{G}(s)$ denote a Monte-Carlo return estimate from replay. We define a BC-loss filter

$$\text{FILTER}(s, z; \epsilon) = \mathbb{I}\Big[Q_\phi(s, a^{\text{cur}}) \geq Q_\phi(s, a^{\text{pre}}) \wedge$$
$$Q_\phi(s, a^{\text{cur}}) - \widehat{G}(s) \leq \epsilon\Big], \quad (6)$$

where the second condition prevents the actor from exploiting critic overestimation by requiring the predicted value to be consistent with $\widehat{G}(s)$, and $\epsilon$ is a small negative constant used to further guard against overestimation.

We first define a filtered BC-style residual regularizer,

$$\mathcal{L}_{\text{BC}}(\theta) = \frac{1}{K} \sum_{k=1}^K \Big(1 - \text{FILTER}(s_t, z_k; \epsilon)\Big) \|s_\theta(s_t, z_k)\|_2^2. \quad (7)$$

The residual actor is then trained to jointly optimize the RL objective and this regularizer:

$$\mathcal{L}_{\text{actor}}(\theta) = \mathcal{L}_{\text{RL}}(\theta) + \beta \mathcal{L}_{\text{BC}}(\theta). \quad (8)$$

## 5. Experiments

We compare against prior RL finetuning methods (§ 5.1). We then analyze how properties of the pretrained BC policy relate to downstream finetuning performance (§ 5.2). Next, we study how RL finetuning reshapes the pretrained action distribution and how this change relates to policy robustness (§ 5.3). We further demonstrate DICE-RL on a challenging real-robot belt assembly task (§ 5.4) and conclude with ablations of key design choices (§ 5.5).

### 5.1. Comparison to RL Finetuning Algorithms

We compare DICE-RL against prior methods, focusing on approaches that build on pretrained flow-based policies (Fig. 2). Refer to Appendix C for implementation details. We benchmark on Can, Square, Transport, Tool Hang tasks from the Robomimic benchmark (Mandlekar et al., 2021), and report results for both state-based

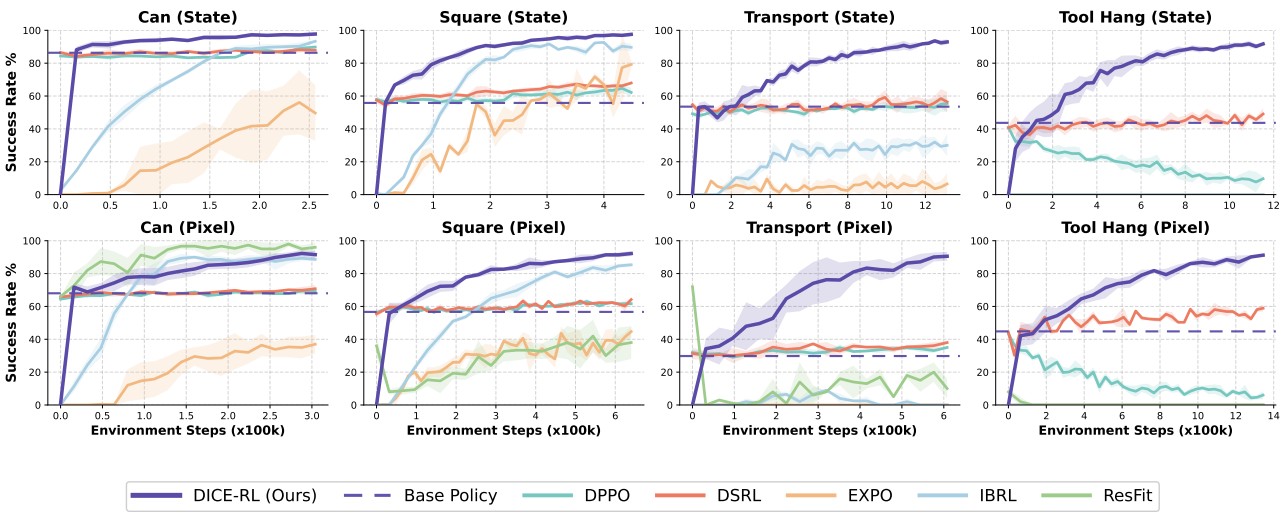

*Figure 2.* **Comparisons on Robomimic.** Success rate versus online environment steps for Robomimic tasks under RL finetuning. Top row: state observations; bottom row: pixel observations. Curves are averaged over 5 random seeds, with evaluation on 300 held-out test configurations; shaded regions denote variability across seeds.

---

**Algorithm 1** DICE-RL

1: **Input:** frozen $\pi_{\text{pre}}$, $D_{\text{demo}}$, chunk horizon $h$, samples $K$, threshold $\epsilon$
2: Init $s_\theta$, critic $Q_\phi$ with target $Q_{\phi'} \leftarrow Q_\phi$, $D_{\text{online}} \leftarrow \emptyset$
3: **for** $t = 1, 2, \ldots, T$ **do**
4:     Observe $s_t$; sample $\{z_k\}_{k=1}^K$
5:     Execute $a_{t:t+h-1} \leftarrow \pi_{\text{pre}}(s_t, z_{k^\star}) + s_\theta(s_t, z_{k^\star})$, $k^\star = \arg\max_k Q_\phi\big(s_t, \pi_{\text{pre}}(s_t, z_k) + s_\theta(s_t, z_k)\big)$
6:     Observe $s_{t+h}$ and $\sum_{j=0}^{h-1} r_{t+j}$; append $(s_t, a_{t:t+h-1}, \sum_{j=0}^{h-1} r_{t+j}, s_{t+h})$ to $D_{\text{online}}$
7:     **if** update step **then**
8:         Sample $B \sim r_{\text{offline}}(t)D_{\text{demo}} + (1 - r_{\text{offline}}(t))D_{\text{online}}$
9:         Update critic toward the multi-sample target $y$ (Eq. (4))
10:        Update residual actor using filtered TD3+BC loss (Eq. (8))
11:        Update targets $\phi' \leftarrow \tau\phi + (1 - \tau)\phi'$
12:     **end if**
13: **end for**

---

and pixel-based observations. We additionally benchmark on LIBERO-10 (Liu et al., 2023); results are provided in Appendix A. For Can, the BC policies are trained on **20** demonstrations; while other tasks using **50** demonstrations from Proficient-Human (PH) dataset. We reduce the number of pretraining demonstrations for two reasons: (i) to leave room for improvement via RL finetuning, and (ii) to better reflect real-world data coverage challenges and constraints. Compared to simulated benchmarks, real-world task distributions are substantially more diverse, and the dynamics can be more complex and stochastic. Achieving a comparable

base-policy success rate in the real world may require collecting substantially more demonstration trajectories than in simulation. We compare against the following baselines:

**IBRL (Hu et al., 2023)** uses online RL on top of pretrained BC policy, by comparing actions from BC and RL policy and picks the one with higher Q-value. IBRL doesn't leverage diffusion-based pretrained policies.

**DPPO (Ren et al., 2024)** finetunes a pre-trained diffusion policy with on-policy policy gradients (PPO) by treating the denoising chain as an inner MDP.

**EXPO (Dong et al., 2025)** finetunes a pretrained diffusion policy by learning a Gaussian edit policy that locally adjusts sampled actions to increase Q-value and with entropy regularization for exploration. Both DICE-RL and EXPO use residual actors, EXPO employs an entropy-regularized Gaussian editor; while we freeze the base and train the residual using a TD3-style Q-maximization objective and a BC regularizer. Since the released EXPO code does not include a pixel-based branch, we adapt it with our vision encoder and similarly freeze visual features during RL finetuning.

**DSRL (Wagenmaker et al., 2025b)** performs RL finetuning in its latent noise space to maximize return, whereas DICE-RL can both optimize the latent noise and apply a learnable residual action correction, which reduces reliance on (and potential bottlenecks from) the pretrained policy. Since the official implementation lacks a pixel-based branch, we wrap the pretrained BC policy with our pretrained checkpoints to support both pixel and state observations. For a fair comparison, we also use the same RLPD-style offline/online mixing schedule as DICE-RL, which empirically improves DSRL's sample efficiency in our setting.

**ResFit (Ankile et al., 2025a)** is an image-based RL finetuning method that freezes a pretrained (action-chunked) BC

policy and learns a lightweight per-timestep residual policy with off-policy actor-critic RL. Unlike DICE-RL, it does not impose an explicit BC-style regularization term or chunked Q learning during finetuning. We include ResFit as a strong baseline in our image-based experiments.

As shown in Fig. 2, DICE-RL attains the highest final performance while also being more stable and sample efficient across all tasks, and it succeeds across all difficulty levels with a single training recipe. ResFit and EXPO are competitive on the easier `Can` and `Square` tasks, but collapse on the more complex long-horizon tasks, potentially due to unbounded exploration and compounding errors in the absence of strong BC regularization. DSRL largely preserves the pretrained policy's initial performance during RL finetuning (avoiding early unlearning), but is less sample efficient than DICE-RL. *To our knowledge, DICE-RL is the first RL finetuning method to reach $\geq 90\%$ success on* `Tool Hang` *from either state or pixel inputs using only 50 demonstrations.* Starting from a `Tool Hang` pretrained BC policy with $45\%$ success, DICE-RL surpasses $90\%$ success within roughly 2000 online episodes.

### 5.2. What makes a pretrained policy easy to finetune?

We ask what properties of a pretrained BC checkpoint make it *easy* to improve with DICE-RL. Because $\pi_{\text{pre}}(s, z)$ is stochastic, finetuning is less about a single "BC action" and more about what the *pretrained action distribution* offers at each state: RL can be sample-efficient when $\pi_{\text{pre}}$ already places non-trivial probability on high-value actions (so finetuning mainly needs to reweight and refine those modes), and when its low-value behavior is *concentrated* rather than diffuse (so the same failure patterns recur and can be corrected reliably by a small residual efficiently).

Guided by this intuition, we evaluate finetunability along three practical pretraining axes, **demonstration quality** (Multi-Human vs. Proficient-Human), **training progress** (checkpoint epoch), and **pretraining data scale** (number of demonstrations), using three complementary metrics: *good mode coverage* (how often $\pi_{\text{pre}}$ already samples near-optimal actions), *bad mode coverage* (how often it does not), and *bad mode entropy* (how diverse the failures are).

Concretely, we use offline demonstration states as anchors and compute the discounted Monte-Carlo return estimate $\widehat{G}(s)$ for each anchor state $s$. We sample $K$ latents and evaluate pretrained candidates $a_k^{\text{pre}} = \pi_{\text{pre}}(s, z_k)$ using the critic $Q_\phi(s, a_k^{\text{pre}})$. We label a pretrained sample as "good" if it passes a threshold relative to the anchor return:

$$\mathbb{I}_{\text{good}}(s, z) \triangleq \mathbb{I}\Big[ Q_\phi\big(s, \pi_{\text{pre}}(s, z)\big) \geq \alpha \, \widehat{G}(s) \Big]. \quad (9)$$

The *good mode coverage* is defined as:

$$\text{GOODCOV}(\pi_{\text{pre}}) \triangleq \mathop{\mathbb{E}}_{\substack{s \sim D_{\text{demo}} \\ z \sim \mathcal{N}(0, I)}} \big[\mathbb{I}_{\text{good}}(s, z)\big], \quad (10)$$

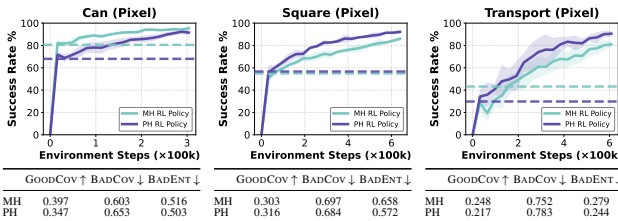

| | GOODCOV↑ BADCOV↓ BADENT↓ | | | GOODCOV↑ BADCOV↓ BADENT↓ | | | GOODCOV↑ BADCOV↓ BADENT↓ | |
|---|---|---|---|---|---|---|---|---|
| MH | 0.397 | 0.603 | 0.516 | MH | 0.303 | 0.697 | 0.658 | MH | 0.248 | 0.752 | 0.279 |
| PH | 0.347 | 0.653 | 0.503 | PH | 0.316 | 0.684 | 0.572 | PH | 0.217 | 0.783 | 0.244 |

*Figure 3.* DICE-RL using either Proficient-Human (PH) or Multi-Human (MH) data. Top: success-rate curves. Bottom: finetunability metrics (GOODCOV/BADCOV/BADENT).

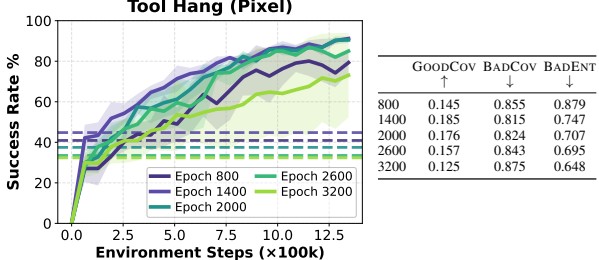

| | GOODCOV ↑ | BADCOV ↓ | BADENT ↓ |
|---|---|---|---|
| 800 | 0.145 | 0.855 | 0.879 |
| 1400 | 0.185 | 0.815 | 0.747 |
| 2000 | 0.176 | 0.824 | 0.707 |
| 2600 | 0.157 | 0.843 | 0.695 |
| 3200 | 0.125 | 0.875 | 0.648 |

*Figure 4.* DICE-RL finetuning with pretrained BC checkpoints trained for different numbers of epochs. Left: success-rate curves. Right: finetunability metrics (GOODCOV/BADCOV/BADENT).

and *bad mode coverage* is defined as $\text{BADCOV}(\pi_{\text{pre}}) = 1 - \text{GOODCOV}(\pi_{\text{pre}})$. Finally, *bad mode entropy* $\text{BADENT}(\pi_{\text{pre}})$ is the average entropy of the "bad" mass: for each anchor state $s$, we draw $K$ latents and keep the action chunks $a = \pi_{\text{pre}}(s, z)$ with $Q_\phi(s, a) < \alpha \widehat{G}(s)$. Each action chunk $a_{t:t+h-1} \in \mathbb{R}^{h \times d}$ (horizon $h$, action dimension $d$) is reshaped into a vector in $\mathbb{R}^{hd}$. We then estimate its entropy by computing a histogram entropy for each coordinate (50 bins over $[-1, 1]$) and averaging over the $hd$ coordinates and across anchor states.

**Demonstration Quality.** We observe differences between pretraining with Proficient-Human (PH) and Multi-Human (MH) when viewed through our finetunability metrics (Fig. 3). Interestingly, for `Can` and `Transport`, MH checkpoints exhibit higher GOODCOV than PH, indicating that the pretrained action distribution more often contains high-value actions; this aligns with its stronger pretrained performance. However, on the `Transport` task, MH shows substantially higher BADENT, meaning its low-value samples are more spread out. This increases the variability of failure modes and makes RL finetuning less consistent and efficient, which correlates with the weaker finetuned performance despite MH's stronger initial performance.

**Convergence Status.** We further probe finetunability by sweeping BC checkpoints across training epochs (Fig. 4). As the BC policy becomes more converged, it assigns more probability mass to action chunks that are worse than what is possible at the same state (lower GOODCOV), while the overall action entropy also shrinks. In this regime, RL

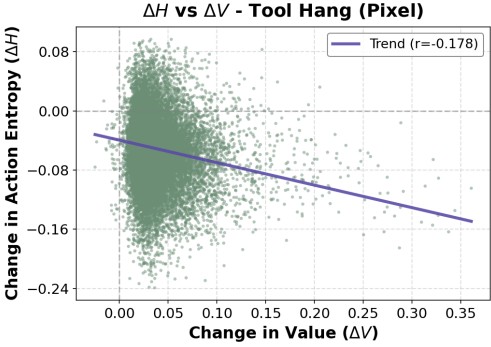

Figure 5. Value improvement ($\Delta V$) vs action entropy reduction ($\Delta H$). Larger gains in value are accompanied by larger drops in action entropy, indicating that RL sharpens the pretrained action distribution to high-value actions.

has fewer value-improving candidates to build on, even though the distribution is more peaked. Empirically, the best finetuning results occur at intermediate checkpoints that balance these two effects: they retain relatively high GOODCOV while already suppressing low-value variability (lower BADENT), whereas later checkpoints trade away too much GOODCOV for additional entropy reduction.

We also vary the number of expert demonstrations used for BC pretraining and find that more demonstrations improve the sample efficiency of RL finetuning; detailed results are provided in Appendix B.

### 5.3. Understanding DICE-RL

This subsection studies *why* DICE-RL improves both performance and robustness. Empirically, we find two complementary effects: (i) RL reshapes the pretrained action distribution to concentrate around high-value actions (*distribution sharpening*), and (ii) the resulting closed-loop behavior becomes more funneling, bringing trajectories from different starting points back together over time (*contraction*). We next validate and quantify both effects.

**Distribution Sharpening.** Our finetuning objective combines critic value maximization with a BC-style residual penalty, and uses a gate to remove this penalty whenever the residual-edited action is confidently value-improving over the pretrained action. This design encourages conservative updates: the policy stays close to the pretrained behavior by default, but preserves value-improving corrections once they are found. As training progresses, this shifts probability mass away from low-value action samples and toward consistently high-value regions, yielding a *sharper* action distribution at visited states.

We test the sharpening hypothesis by jointly measuring value improvement and action entropy after finetuning. Using states from the offline demonstrations $D_{\text{demo}}$ as anchors,

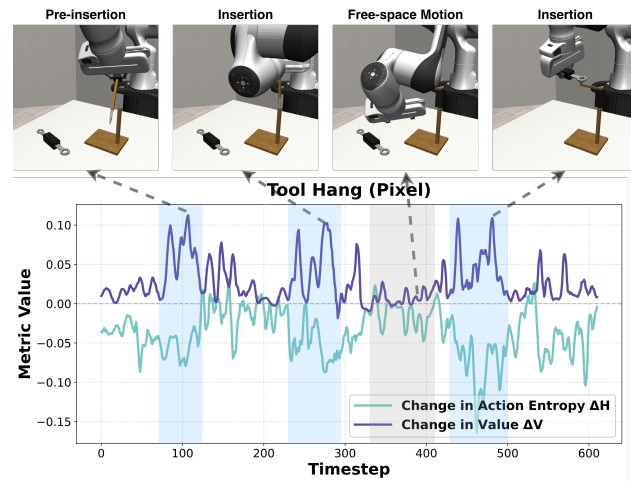

Figure 6. A representative rollout trajectory of the RL policy on `Tool Hang`, together with the running change in action entropy and value improvement. We zoom in on frames where value improvement spikes and action entropy drops; these states are often critical for task success (e.g., pre-insertion and insertion). In contrast, during free-space motions that are less consequential for success, we observe less reduction in action entropy

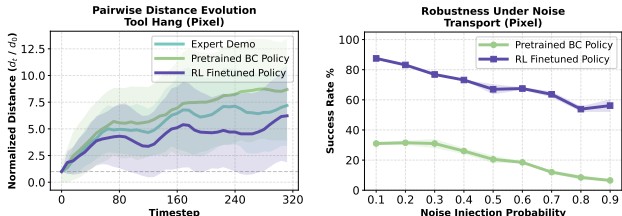

Figure 7. (Left) Pairwise distance over time between two rollouts from different initial states, $\frac{|s_t - s'_t|_2^2}{|s_0 - s'_0|_2^2}$, comparing our RL policy to expert demonstrations. The rollouts from RL policy exhibit more stable distance evolution, indicating stronger contraction. (Right) Success rate under action noise injection, where the probability of perturbing the action output increases from 0.1 to 0.9. The RL policy degrades more gracefully than the pretrained BC policy.

we sample actions from the finetuned policy and compute (i) value gain relative to the pretrained BC policy and (ii) the empirical entropy drop of the sampled actions. We find a clear coupling: states with larger value improvements exhibit larger entropy drops, suggesting that successful finetuning coincides with stronger distributional concentration (Fig. 5). This effect is most pronounced at *critical* states along the trajectory. In a representative `Tool Hang` rollout ((Fig. 6)), entropy decreases around contact-rich phases such as pre-insertion and insertion, precisely where value rises the most, while entropy remains comparatively stable during free-space motions that tolerate lower precision.

**Contraction and Policy Robustness.**

Distribution sharpening is a *state-local* effect: conditioned on a fixed state, RL concentrates the policy's action distribution toward higher-value regions (e.g., reduced entropy

or variance). Contraction, in contrast, is a *trajectory-level* property of the *closed-loop dynamics induced by a policy*: over a task-relevant region and in a chosen metric, trajectories initialized from nearby states move closer over time, reflecting reduced sensitivity to initial conditions. This notion is closely related to *incremental stability* and connects to robust-control "funnel" or trajectory-tube intuitions (Lozano-Perez et al., 1984; Tsukamoto et al., 2021), where stability of a tube around nominal behavior implies improved robustness to perturbations.

To probe contraction empirically, we sample many pairs of nearby anchor states $(s_0, s_0')$ from the offline demonstrations $D_{\text{demo}}$. From each pair, we rollout (i) the fine-tuned RL policy for $T$ steps to obtain $\{s_t^{\text{RL}}\}_{t=0}^T$ and $\{s_t'^{\text{RL}}\}_{t=0}^T$, (ii) the pretrained BC policy for $T$ steps to obtain $\{s_t^{\text{Pre}}\}_{t=0}^T$ and $\{s_t'^{\text{Pre}}\}_{t=0}^T$, and (iii) the corresponding expert trajectories of length $T$ starting from the same anchors, denoted $\{s_t^{\text{E}}\}_{t=0}^T$ and $\{s_t'^{\text{E}}\}_{t=0}^T$. We then measure the normalized pairwise divergence for each rollout type $x \in \{\text{RL}, \text{Pre}, \text{E}\}$:

$$c^x(t) = \frac{\left\| s_t^x - s_t'^x \right\|_2^2}{\left\| s_0 - s_0' \right\|_2^2}. \tag{11}$$

where $s_t^x$ denotes the encoded feature of the high-dimensional observation (we write $s_t^x$ for brevity) and distances are computed in this latent space. As shown in Fig. 7 (Left), rollouts under our RL policy exhibit a more stable (and typically smaller) evolution of $c(t)$ than both the pretrained BC policy and the expert demonstration rollouts, indicating stronger contraction of the closed-loop behavior.

Contraction suggests improved robustness: if trajectories are less sensitive to state perturbations, then moderate disturbances should be attenuated rather than amplified. We test this implication by injecting action noise with increasing probability (from 0.1 to 0.9) during execution and measuring success rate. Fig. 7 (Right) shows that the RL policy degrades more gracefully under perturbations than the pretrained BC policy, consistent with the stronger contraction observed in the trajectory-divergence analysis. We do not claim that RL finetuning improves out-of-distribution generalization beyond the pretrained BC policy (the finetuned RL policy should be at least as generalizable as the BC policy). Rather, our results indicate improved robustness to small, on-trajectory perturbations.

### 5.4. DICE-RL on Real Robot

We evaluate DICE-RL on three challenging real robot tasks: `GearInsertion`, `LightBulbInsertion`, and `BeltAssembly` (from the NIST benchmark (NIST); Fig. 8). All three tasks are contact-rich and require high action precision.

In `GearInsertion`, the robot needs to insert a gear onto a metal rod. This task has a tight error toler-

ance ($\approx 1$ mm) and also involves visual occlusion. In `LightBulbInsertion`, the robot needs to first insert the light bulb into the socket, then rotate it by about 360 degrees to turn it on. `BeltAssembly` requires precise coordination with both rigid and deformable objects: starting with a rubber belt already grasped, the robot has to thread the belt around two pulleys. Minor errors can cause the belt to slip, snag, or snap off a pulley during threading, making the task highly sensitive to contact interactions.

**BC Pretraining.** We collect expert demonstrations using a kinesthetic teaching interface and train a base diffusion policy for each task. For `GearInsertion`, `LightBulbInsertion`, and `BeltAssembly`, we collect 40, 100, and 265 demonstration trajectories, respectively. Refer to Appendix D for implementation details.

Although the pretrained policies exhibit reasonable behavior, they still show several dominant failure modes (Fig. 8(a)). For example, in `BeltAssembly`, the BC policy is prone to (i) slipping off the first pulley when transitioning too aggressively to the second, (ii) failing to properly seat the belt on the second pulley, and (iii) releasing the belt prematurely. In `GearInsertion` and `LightBulbInsertion`, the most common failure modes are (i) poor navigation when approaching the target objects and (ii) imprecise insertion into the target.

**Real Robot RL Setup.** Our real robot RL setup involves a human operator to supervise rollouts, provide binary success/failure rewards, and perform resets. The operator terminates trajectories upon clear failure and assigns reward 0; otherwise, they assign reward 1 at the end of the episode. In practice, consistent labeling across successes and failures is important for stable value learning and policy updates. To reduce wall-clock overhead, we do batched RL updates after every 10 online episodes. The environment runner (policy actor) and RL learner run as separate processes (Luo et al., 2024; Ankile et al., 2025a): the learner loads the latest episodes, adds them to the replay buffer, and updates the networks asynchronously.

**Evaluation.** We evaluate RL checkpoints at fixed training intervals using 30 trials per checkpoint. For `GearInsertion` and `LightBulbInsertion`, we keep the target object poses fixed and randomize the robot end-effector's initial pose within a small range (approximately $\pm 2$ cm in position and $\pm 10°$ about each of the three rotation axes). For `BeltAssembly`, in addition to randomizing the robot's initial pose, we also randomize the assembly board position within approximately $\pm 10$ cm and its planar orientation within approximately $\pm 15°$. The evaluation curves over the course of RL finetuning are shown in Fig. 8(c). Fig. 9 shows a representative `BeltAssembly` rollout annotated with value improvement and the running change in action entropy. Notably, the largest value gains

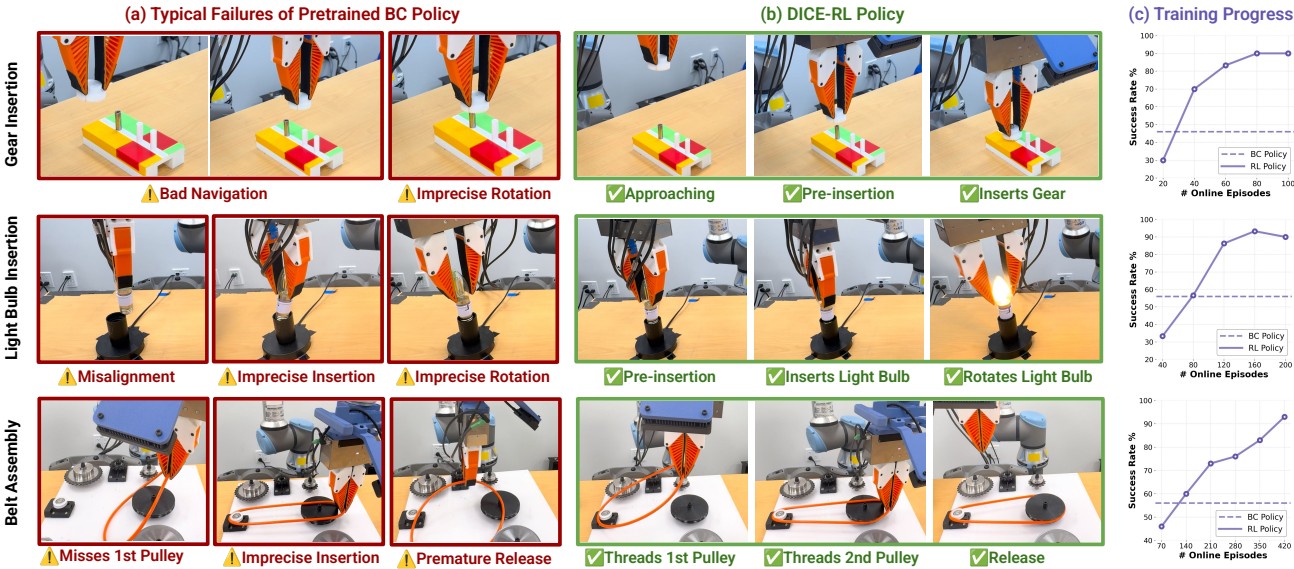

*Figure 8.* **Real robot experiment.** **(a)** The dominant failure modes of the pretrained BC policies for all three tasks. **(b)** After RL finetuning, DICE-RL works reliably on all tasks. **(c)** Success rate versus number of online episodes.

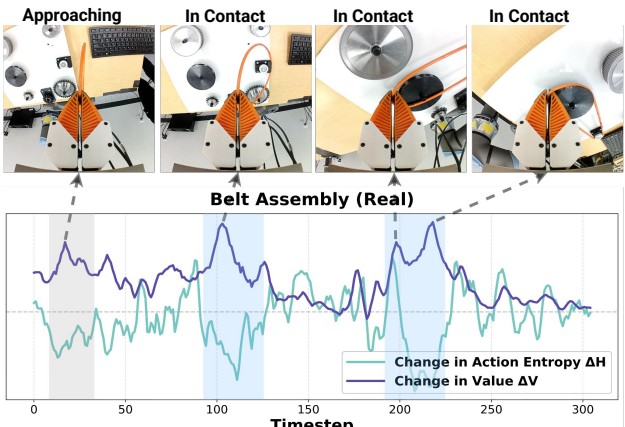

*Figure 9.* A representative `BeltAssembly` rollout with running change in action entropy ($\Delta H$) and value improvement ($\Delta V$), with the largest entropy drop and value gains occurring around critical contact transitions.

and entropy reductions occur at states corresponding to the BC policy's dominant failure modes (e.g., pulley transitions and belt seating), consistent with the distribution-sharpening effects we observed in simulation experiments.

### 5.5. Ablation Studies

We ablate three key design choices in DICE-RL: (i) the BC-loss filter, (ii) multi-sample expectation training, and (iii) best-of-$N$ action selection. All three generally speed up convergence and improve peak RL finetuning performance. We also vary the pretrained policy backbone (diffusion vs. flow matching) and find that DICE-RL finetunes stably and sample efficiently for both pretrained policy architectures. Additional ablations on these design decisions and key hy-

perparameters are provided in Appendix B.

## 6. Conclusion

We present DICE-RL, a stable and sample-efficient RL fine-tuning framework for generative behavior cloning policies. Beyond empirical gains, our analyses show that RL fine-tuning systematically sharpens and contracts the pretrained action distribution around high-value behaviors. DICE-RL achieves strong performance across tasks and observation modalities, including a challenging real-world belt assembly task. Future work could extend DICE-RL to finetune large multi-task VLA policies that exhibit richer, more diverse exploration, with the goal of simultaneously preserving behavioral diversity while achieving high-precision control; it will also be interesting to develop a deeper theoretical understanding of RL finetuning for generative policies, including what guarantees can be obtained for stability and sample efficiency.

## Acknowledgment

This work was supported in part by the NSF Award #2143601, #2037101, and #2132519, and Toyota Research Institute. We would like to thank TRI for the UR5 robot hardware. The views and conclusions contained herein are those of the authors and should not be interpreted as necessarily representing the official policies, either expressed or implied, of the sponsors. We thank all REALab members for providing value feedback on an early draft of this work. We are also grateful to Somil Bansal, Perry Dong, Masha Itkina, Qiyang (Colin) Li, Paarth Shah, Max Simchowitz, Chen Xu, and Wenxuan Zhou for thoughtful discussions.

## Impact Statement

This work improves the stability and sample efficiency of RL finetuning for pretrained generative robot policies and validates the approach in simulation and on a real robot. As with other BC and RL-based methods, the resulting policies may inherit dataset biases or exhibit unsafe behavior under distribution shift, and should be carefully evaluated with appropriate safeguards before deployment in safety-critical settings. We do not anticipate additional societal impacts that require specific discussion.

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

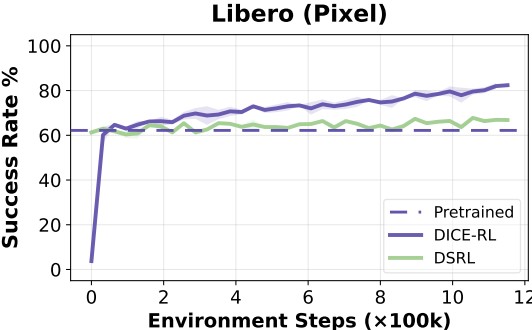

*Figure 10.* DICE-RL on LIBERO-10, averaged across all 10 tasks with 100 rollouts per task.

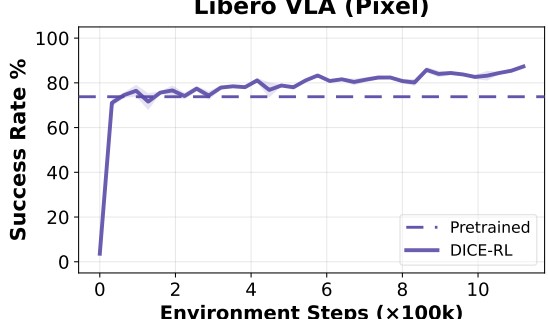

*Figure 11.* DICE-RL with $\pi_0$ on LIBERO-10.

# Appendix

## A. Multitask and VLA Result

**DICE-RL on LIBERO-10.** To validate RL finetuning on a multitask manipulation benchmark, we apply DICE-RL to LIBERO-10 (Liu et al., 2023). We first train a multitask BC policy on 500 demonstrations (50 per task) and then finetune it with DICE-RL.During online finetuning, we randomly sample a task at each iteration for rollout collection. As shown in Fig. 10, DICE-RL outperforms DSRL in sample efficiency. Fig. 12 provides a per-task breakdown.

**DICE-RL with $\pi_0$ on LIBERO-10.** We further integrate DICE-RL with the pretrained $\pi_0$ model (Black et al., 2024). Starting from the pretrained checkpoint, we first perform supervised finetuning with 30 demonstrations per task, followed by RL finetuning. $\pi_0$ processes an observation in two stages: it first encodes the camera images and language instruction, then generates actions via iterative denoising. To obtain a latent observation, we reuse only the first stage: the images (through $\pi_0$'s vision encoder) and the instruction (through its text embedder) are fed into $\pi_0$'s transformer for a single forward pass that stops before action generation, so the resulting latent is action-free and does not require iterative denoising. We average the transformer out-

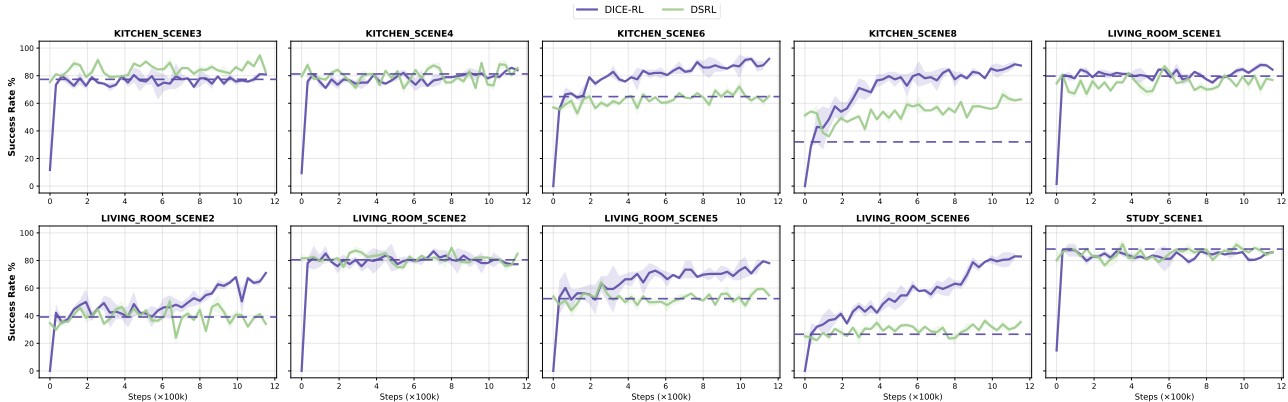

*Figure 12.* Per-task breakdown of DICE-RL on LIBERO-10.

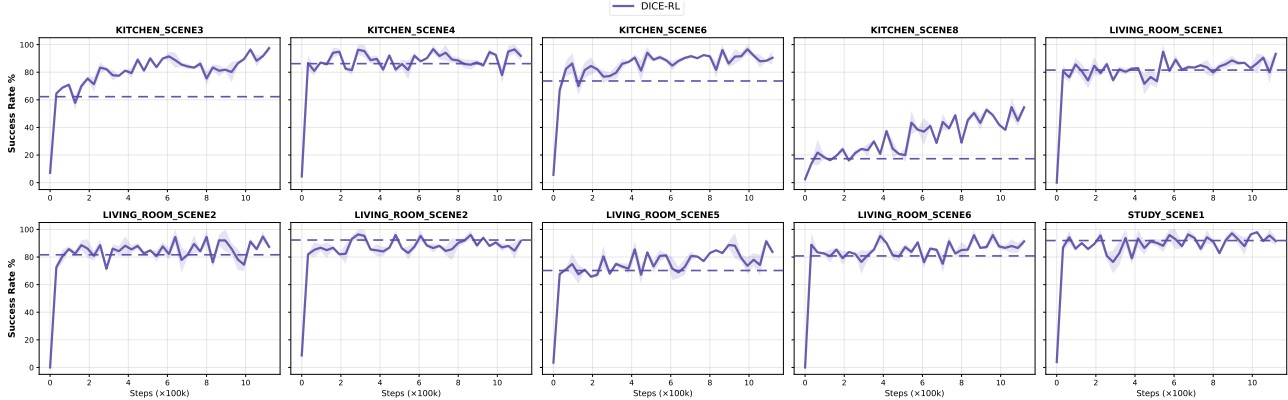

*Figure 13.* Per-task breakdown of DICE-RL with $\pi_0$ on LIBERO-10.

put across all image and text token positions into a single 2048-dimensional vector serving as the latent observation. The base action is obtained separately from $\pi_0$'s ordinary full inference; the residual actor learns a correction on top of it. The latent observation is cached and shared between the residual actor and critic. As shown in Fig. 11, DICE-RL stably improves online performance. Fig. 13 provides a per-task breakdown.

## B. Detailed Ablations Results

In this section, we present detailed ablations of key design choices in DICE-RL and the choice of pretrained policy architecture. We also study the effect of the number of demonstrations used for pretraining. Finally, we conclude with ablations on several key RL finetuning hyperparameters. All results are averaged across 5 random seeds.

**BC Loss Filter.** We ablate the BC loss filter, which conditionally disables the BC regularizer when the filter criterion is triggered (Eq. (6)). As shown in Fig. 14, the filter consistently improves sample efficiency and final performance on most tasks, with the largest gains observed on the long-horizon `Transport` and `Tool Hang` tasks.

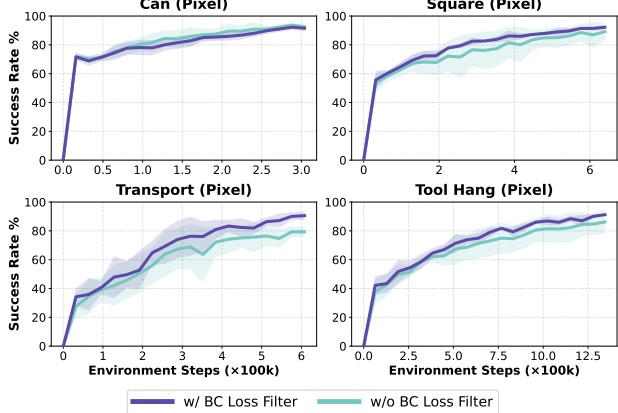

*Figure 14.* The BC-loss filter prevents RL finetuning from being bottlenecked by the pretrained policy.

**Multi-sample Training.** We ablate the multi-sample expectation training strategy from Sec. 4 by varying the number of action samples used state, $K \in 1, 4, 16$. Instead of optimizing the actor (and bootstrapping the critic) using a single sampled action chunk, we average objectives over $K$ latent-induced candidates, allowing the residual to improve the *entire* action distribution induced by $\pi_{\text{pre}}(s, z)$ while reduc-

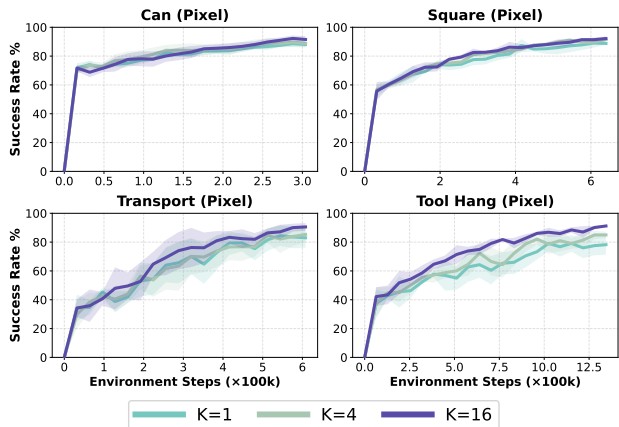

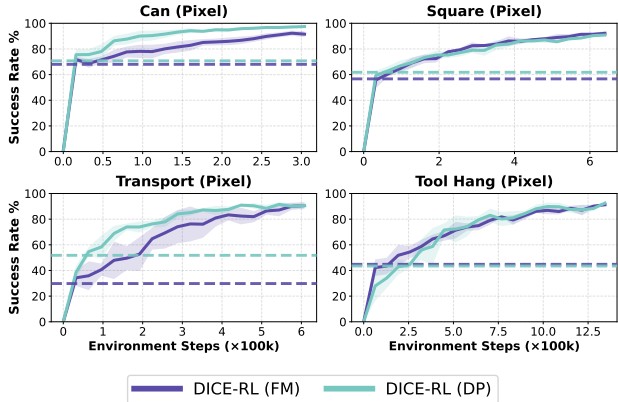

*Figure 17.* DICE-RL achieves stable, sample-efficient RL finetuning for both pretrained flow policies and diffusion policies.

*Figure 15.* Multi-sample expectation training improves RL finetuning sample efficiency, with larger $K$ yielding faster gains.

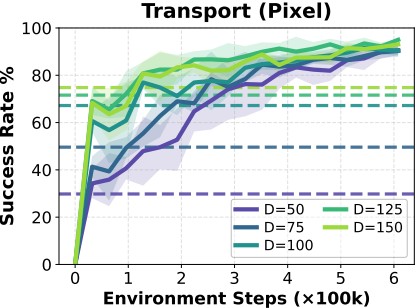

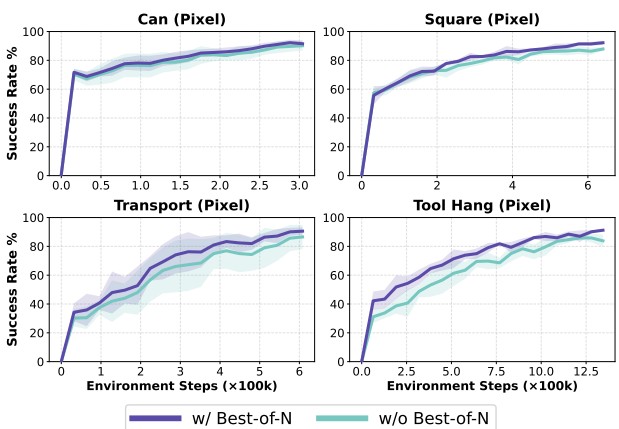

*Figure 18.* DICE-RL finetuning with varying number of demonstrations used for BC policy pretraining.

*Figure 16.* Best-of-$N$ action selection during online interaction accelerates RL finetuning convergence to peak performance. Dashed lines indicate performance of pretrained BC policies.

ing gradient variance through sample reuse. As shown in Fig. 15, increasing $K$ consistently improves sample efficiency, highlighting the benefit of distribution-level policy optimization.

**Best-of-$N$ Action Selection.** We assess best-of-$N$ action selection at online interaction time, where we sample $K$ latent-conditioned action chunks and execute the candidate with the highest predicted value under $Q_\phi$. This provides a lightweight performance boost by exploiting the latent-induced diversity of the pretrained policy without changing the training procedure (Fig. 16). Notably, even without best-of-$N$, DICE-RL still attains strong performance on all four tasks, while best-of-$N$ further accelerates convergence and improves peak performance.

**Pretrained Policy Architecture.** While most experiments use a flow matching policy as the pretrained BC backbone, we also evaluate DICE-RL with a diffusion policy back-

bone. To isolate the effect of the pretraining objective, we keep the observation encoder and denoising network architectures identical and train the BC policy using either a flow-matching loss or a DDPM loss. We refer to DICE-RL with the diffusion-pretrained BC policy as DICE-RL (DP), and the default flow-matching-pretrained version as DICE-RL (FM). For DICE-RL (DP), we perform RL finetuning using DDIM sampling with $\eta = 0$ ensure determinism. Fig. 17 shows that both DICE-RL (FM) and DICE-RL (DP) finetune stably and sample efficiently, reaching comparable peak performance.

**Varying the Number of Pretraining Demonstrations.** We ablate the amount of offline data used for BC pretraining. On `Transport`, we pretrain the BC policy using $\{50, 75, 100, 125, 150\}$ demonstrations from the Proficient-Human dataset. As shown in Fig. 18, increasing the number of demonstrations empirically improves downstream RL finetuning, yielding better sample efficiency and higher peak performance.

**Number of Gradient Updates** We ablate the number of gradient steps per RL update, which is closely related to the update-to-data (UTD) ratio. When the RL update frequency

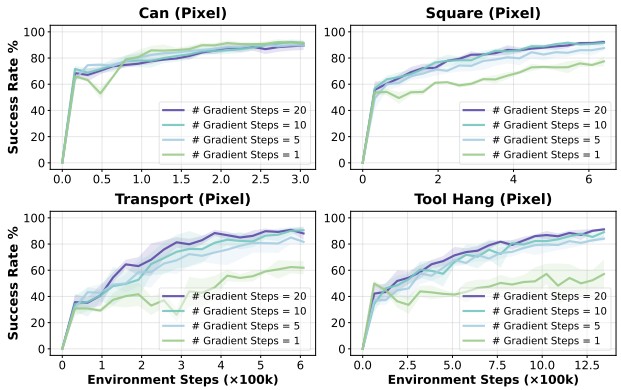

*Figure 19.* DICE-RL finetuning with varying number of gradient steps per RL update.

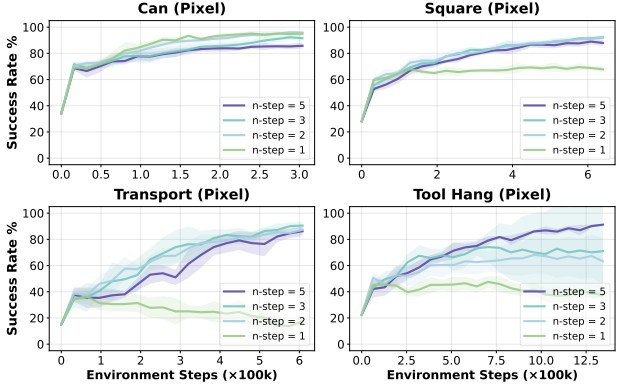

*Figure 20.* DICE-RL finetuning with varying number of N-step return.

is fixed, increasing the number of gradient steps per agent update corresponds to a higher UTD ratio. As shown in Fig. 19, a higher UTD ratio is most beneficial for long-horizon tasks. On `Can`, however, finetuning performance drops slightly as the UTD ratio increases. We hypothesize that this is because, in long horizon tasks, each collected transition carries a more delayed learning signal, so a larger UTD ratio improves sample efficiency by extracting more value from rare informative trajectories. In contrast, shorter horizon tasks tend to provide denser signals with easier credit assignment, which are already captured with relatively few updates; additional updates may then lead to overfitting or over-optimization.

**N-step Return** We next ablate the choice of n in n-step return. This hyperparameter controls the temporal span used in the TD target: instead of bootstrapping after one step ($n{=}1$), the critic uses $n$ steps of real rewards before bootstrapping, which improves credit propagation but can also increase target variance and off-policy mismatch. Empirically, from Fig. 20, we observe that $n{=}1$ works best for the shortest-horizon task, `Can`, while increasing $n$ hurts per-

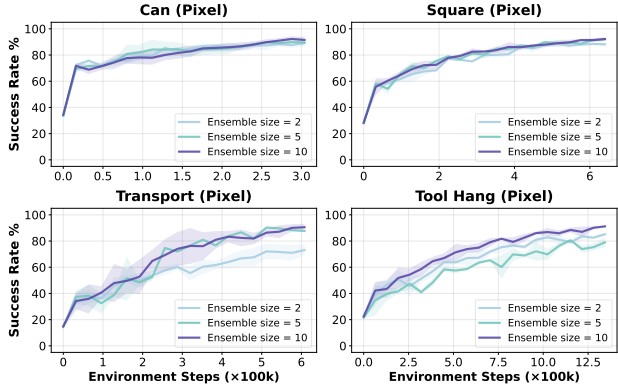

*Figure 21.* DICE-RL finetuning with varying number of critic networks used in the critic ensemble.

formance there; for longer-horizon tasks, $n{=}3$ improves sample efficiency; and for two tasks with similar nominal horizon, `Transport` and `Tool Hang`, the more precision-demanding task `Tool Hang` benefits more from a even larger value ($n{=}5 > 3$), whereas the less precision-demanding one prefers a smaller value ($n{=}3$). A plausible explanation is that the difference is not just from horizon or whether a task has failure-critical phases, but how *strict* and *stacked* those phases are. In the more precise task, there are more points where the policy must be correct within a tighter tolerance, so the final outcome depends more strongly on whether several earlier fine-grained decisions were all set up correctly. As a result, the payoff of early actions is revealed over a longer phase of downstream interactions, and a larger $n$ can better propagate these delayed outcomes back to the earlier actions; in easier or shorter tasks, this dependency chain is weaker, so smaller $n$ is often enough.

**Critic Ensemble Size** is the number of Q-functions used to estimate value, where the value network update uses the minimum across the ensemble to reduce overestimation. As shown in Fig. 21, empirically, increasing the ensemble size from the default of 2 to 5 or 10 improves sample efficiency on the two longer horizon tasks, `Transport` and `Tool Hang`, while the benefit is less pronounced on the two shorter-horizon tasks; but larger ensembles do not seem to hurt performance. A plausible explanation is that larger ensembles provide a more stable and conservative value signal by averaging out critic errors before the min-reduction, which is especially helpful in long-horizon tasks where small value errors can compound through many backup steps and mislead policy improvement. In shorter horizon tasks, value estimation is less fragile to such compounding errors, so the marginal gain from additional critics is smaller.

**BC Loss Weight** We ablate the effect of the BC regularization loss coefficient, i.e., $\beta$ in Eq. 8. As shown in Fig. 22, we find that $\beta{=}50$ performs best on `Can` and `Square`,

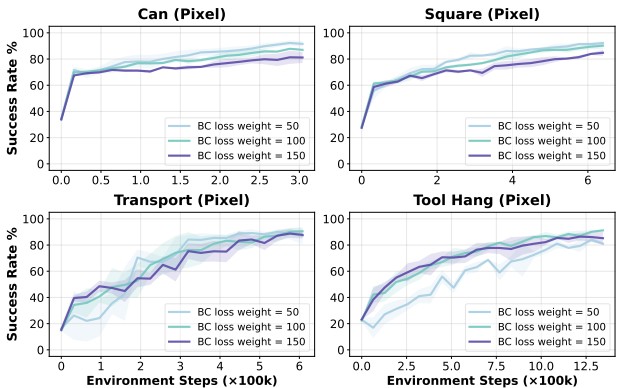

*Figure 22.* DICE-RL finetuning with varying BC loss coefficients.

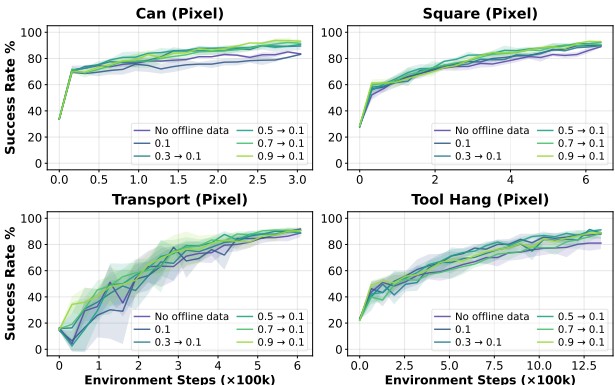

*Figure 23.* DICE-RL finetuning with varying RLPD schedule.

while increasing the weight to 100 or 150 degrades finetuning performance. On `Transport`, all three values have comparable results. For `Tool Hang`, $\beta$=100 achieves the best performance. These trends suggest that the optimal BC weight might depend on how much RL improvement requires deviating from the pretrained action distribution. On shorter-horizon tasks (`Can`, `Square`), a moderate BC weight ($\beta$=50) is sufficient to stabilize learning; stronger BC regularization can over-constrain the exploration and suppress the larger corrections needed for improvement, leading to worse final performance. In contrast, for the long-horizon, high-precision task (`Tool Hang`), a larger $\beta$ (100) is beneficial: stronger BC regularization helps prevent destabilizing drift, and in high-precision regimes the desired residual corrections are often small in magnitude, so they are less penalized by a larger BC loss.

**RLPD Schedule** We ablate different RLPD schedules as described in Sec. 4. We compare several decay schedules, a fixed offline-data ratio of 0.1, and a variant that removes offline demonstrations entirely during online finetuning (Fig. 23). Interestingly, for the easier, shorter-horizon tasks `Can` and `Square`, using a higher initial fraction of offline data yields better finetuning performance, whereas

the trends are less conclusive on `Transport` and `Tool Hang`. On `Transport`, retaining more offline data accelerates the early warmup phase of the residual policy, but the final converged performance differs only marginally across schedules. On `Tool Hang`, removing offline data during online finetuning degrades the final performance more noticeably, while the choice among the remaining schedules has a relatively small effect.

Overall, once a reasonable baseline configuration is chosen, finetuning performance is not particularly sensitive to modest changes in any single hyperparameter. This relative insensitivity suggests that our framework does not rely on brittle tuning of individual knobs, and the ablation results collectively indicate that the proposed finetuning procedure is robust across different tasks from Robomimic.

## C. Implementation Details - Simulation

We summarize implementation details for BC pretraining and RL finetuning for the results in Sec. 5.1.

**BC pretraining.** For state observations, we use a lightweight MLP encoder over the concatenated proprioceptive state. For pixel observations, we use a multi-view ResNet-18 encoder: each camera view is processed by an independent ResNet-18 trained from scratch (no ImageNet pretraining). Each encoder takes stacked RGB frames of shape $(3, H, W)$. We apply random cropping as data augmentation during training and center cropping at evaluation. Crop settings are task-dependent: `Tool Hang` uses $216\times216$ crops from $240\times240$ images; `Square` and `Can` use $84\times84$ crops from $96\times96$ images; and `Transport` uses the full $96\times96$ images without cropping. Multi-view features are concatenated across views, then concatenated with the (flattened) proprioceptive state and passed through a linear projection to form the conditioning input to the policy. We pretrain a conditional flow-matching policy over action chunks. The flow network is a 1D U-Net operating along the action horizon, with channel multipliers $\{1, 2, 2\}$ and base width 128. Each resolution level contains residual blocks with FiLM conditioning on time and the state features. Training follows the standard linear-path flow-matching objective: given a data action chunk $x_1$ and Gaussian noise $x_0 \sim \mathcal{N}(0, I)$, we sample $t \sim \text{Unif}[0, 1]$ and form

$$x_t = (1 - t)x_0 + tx_1. \tag{12}$$

The model predicts a conditional velocity field $v_\theta(x_t, t, c)$ and is trained with

$$\mathcal{L}_{\text{FM}} = \mathbb{E}_{x_1 \sim D_{\text{demo}}, x_0, t}\left[\left\|v_\theta(x_t, t, c) - (x_1 - x_0)\right\|_2^2\right], \tag{13}$$

where $c$ denotes the encoder features (and optionally proprioception). We discretize the ODE time interval $t \in [0, 1]$ into `flow_steps` steps and use the same forward-Euler integrator during both training (for sampling action chunks from

*Table 1.* Hyperparameters for BC Pretraining across Robomimic tasks

| Hyperparameter | Transport | Tool Hang | Square | Can |
|---|---|---|---|---|
| Proprioceptive dim | 18 | 9 | 9 | 9 |
| Action dim | 14 | 7 | 7 | 7 |
| # cameras | 4 | 2 | 2 | 2 |
| Image res. | 96×96 | 240×240 | 96×96 | 96×96 |
| Crop size | None | 216×216 | 84×84 | 84×84 |
| Flow steps | 10 | 20 | 10 | 10 |
| Horizon steps | 8 | 8 | 4 | 4 |

*Table 3.* Hyperparameters for BC Pretraining across real robot tasks

| Hyperparameter | GearInsertion | LightBulbInsertion | BeltAssembly | |
|---|---|---|---|---|
| Proprioceptive dim | 9 | 9 | 9 | |
| Action dim | 9 | 10 | 9 | |
| # cameras | 4 | 2 | 2 | 2 |
| Image res. | 224×224 | 224×224 | 224×224 | |
| DDIM steps | 8 | 12 | 16 | |
| Horizon steps | 12 | 16 | 16 | |

the model) and inference. We set `flow_steps=10` for `Transport`, `Square`, and `Can`, and `flow_steps=20` for `Tool Hang` due to its higher precision requirements. Tab. 1 summarizes hyperparameters for BC training.

**RL Finetuning.** We finetune using an off-policy actor–critic over $h$-step action chunks (Sec. 4). The actor is the lightweight residual $s_\theta(s, z)$, while the pretrained generative policy $\pi_{\text{pre}}$ (including the observation encoder) is kept frozen throughout RL. We use an ensemble of $N_Q$ critics to estimate $Q(s, a_{t:t+h-1})$ and mitigate overestimation; all networks use GELU activations with MLP widths [1024,1024,1024] and are trained with Adam at learning rate $10^{-4}$, Polyak target updates with $\tau = 0.01$, and task-specific $n$-step returns. Each online interaction step collects experience from multiple parallel environments, and we perform batched gradient updates with a fixed update-to-data ratio (Tab. 2). Empirically, we find that larger update-to-data ratios (10–20 gradient steps per environment step) improve sample efficiency, larger critic ensembles (e.g., $N_Q$=10 vs. 2) improve stability and final performance, and using multi-step ($n$-step return) TD targets is consistently beneficial on long-horizon tasks.

*Table 2.* Hyperparameters for RL finetuning across Robomimic tasks

| Hyperparameter | Transport | Can | Square | Tool Hang |
|---|---|---|---|---|
| Num of parallel envs | 4 | 4 | 8 | 8 |
| Critic ensemble size | 10 | 10 | 10 | 10 |
| Actor hidden dims | [1024,1024,1024] | [1024,1024,1024] | [1024,1024,1024] | [1024,1024,1024] |
| Critic hidden dims | [1024,1024,1024] | [1024,1024,1024] | [1024,1024,1024] | [1024,1024,1024] |
| LR | 1e-4 | 1e-4 | 1e-4 | 1e-4 |
| Activation | GELU | GELU | GELU | GELU |
| Target update $\tau$ | 0.01 | 0.01 | 0.01 | 0.01 |
| Gradient steps / update | 10 | 10 | 20 | 20 |
| $r_{\text{offline}}^{\text{start}} \rightarrow r_{\text{offline}}^{\text{end}}$ | 0.9→0.1 | 0.7→0.1 | 0.5→0.1 | 0.5→0.1 |
| $T_{\text{ratio}}$ | 320000 | 160000 | 320000 | 640000 |
| $n$-step return | 3 | 3 | 3 | 5 |
| BC loss weight $\beta$ | 100 | 50 | 50 | 100 |
| Q underest. thresh. $\epsilon$ | -0.5 | -0.5 | -0.25 | -0.75 |
| Multi-sample training $K$ | 16 | 16 | 16 | 16 |

## D. Implementation Details - Real Robot

**BC pretraining.** The BC policy is implemented as a diffusion policy that uses a Vision Transformer encoder (ViT-Base with 32×32 patch size, pretrained with CLIP) to extract 768-dimensional feature representations from 224×224 RGB images. The denoising network employs a conditional U-Net architecture with channel dimensions of [256, 512,

1024] and 5×5 convolutional kernels, which iteratively refines action predictions conditioned on the visual features. For data processing, robot state observations are collected at 500Hz and downsampled by a factor of 5, while RGB images are captured at 60Hz and downsampled by a factor of 10. Observations are formed by temporally aligning these multi-rate sensors for each RGB frame, the policy selects the robot state with the closest timestamp, ensuring synchronized visual and proprioceptive inputs. The policy generates action chunks by predicting sequences at 1/50th of the original control frequency. With a prediction horizon of 8 timesteps, each action chunk spans 800ms of robot execution (8 timesteps × 50 downsampling × 2ms per control cycle). Tab. 3 summarizes hyperparameters for BC training.

**RL Finetuning.** The residual actor and critic networks are implemented as MLPs with hidden layers [1024, 1024, 1024], matching the simulated experiments. We use a critic ensemble size of 5 and set the BC loss weight to 100. For the RLPD schedule, we linearly decrease the ratio from 0.5 to 0.1 over the first 100 online episodes. On real-robot tasks, we perform batched online updates of the RL agent and run updates asynchronously once every 10 online episodes. We choose the number of gradient steps such that the total number of gradient updates is approximately matched to the number of transition pairs in the replay buffer (roughly 1:1). Accordingly, we use 2000, 3000, and 3000 gradient steps for `GearInsertion`, `LightBulbInsertion`, and `BeltAssembly`, respectively (varying with task horizon). We use a batch size of 256 for all tasks.

