# OpenReview forum: "From Prior to Pro: Efficient Skill Mastery via Distribution Contractive RL Finetuning"
_ICML.cc/2026/Conference — ICML 2026 regular_

### Official Review · Reviewer_N9LR · 2026-03-10

**Soundness:** 3
**Presentation:** 2
**Significance:** 3
**Originality:** 2
**Overall Recommendation:** 5
**Confidence:** 4

**Summary:**

This paper introduces Distribution Contractive Reinforcement Learning (DICE-RL), an offline-to-online RL framework built upon a pretrained diffusion/flow model policy. It fine-tunes the residual policy in a sample-efficient manner using proper regularization and value guidance. The training pipeline is meticulously designed, incorporating techniques such as Adaptive RLPD mixing and a BC loss filter. The performance of this method is consistently robust across both simulation and real-world robot tasks. Furthermore, the paper presents several interesting findings, such as the "Contraction" phenomenon: the BC policy sharpens during the RL process, suggesting that it is advantageous to use a pretrained policy that has not been fully trained to convergence.

**Compliance With Llm Reviewing Policy:**

Affirmed.

**Final Justification:**

My main concerns have been addressed so I raised my score.

**Key Questions For Authors:**

- In Fig.2, I noticed many baselines method fails in some tasks. Can you explain the failure mode in detail, tell which different design that makes DICE-RL work.

**Limitations:**

The authors have not discussed the limitations.

**Strengths And Weaknesses:**

Soundness:
The paper demonstrates a robust offline-to-online RL framework, DICE-RL, for fine-tuning pretrained diffusion and flow-based policies. The proposed training pipeline, which incorporates techniques like Adaptive RLPD mixing and BC loss filtering, appears technically sound and meticulously engineered. The empirical results across both simulation and challenging real-world robotic tasks convincingly validate the sample efficiency and stability of the method. Furthermore, the analysis regarding the "contraction" phenomenon is insightful. I encourage the authors to provide more detailed ablation studies towards these phenomenon.

Presentation:
The paper is well-organized and clearly written.

Significance:
This paper significantly advances our practical and theoretical understanding of offline-to-online RL fine-tuning for generative behavior cloning policies. By achieving high sample efficiency and stability, it addresses a major bottleneck in robot learning. I encourage the authors to delve deeper into the broader landscape of foundation models and explore the extension of this framework to large multi-task Vision-Language-Action (VLA) policies, as briefly mentioned in their conclusion.

Originality:
This paper provides practical RL fine-tuning method for generative policies. The carefully designed pipeline distinguishes this work from existing literature in the offline-to-online RL domain.

---

> ### Author Rebuttal · Authors · 2026-03-31
>
> > In Fig.2, I noticed many baseline methods fail in some tasks. Can you explain the failure mode in detail, tell which different design makes DICE-RL work.
>
> We thank the reviewer for this insightful comment. Below, we provide a clearer explanation of why DICE-RL outperforms each baseline.
>
> - ResFiT: ResFiT is weaker mainly for two reasons: (1) it performs per-step residual action learning and value learning, whereas DICE-RL performs chunk-level residual RL; and (2) it does not use an explicit BC regularization term during finetuning. Our new ablation at https://dice-rl-anonymous.github.io/#rebuttal-resfit isolates these two differences cleanly by comparing **DICE-RL, DICE-RL w/o BC loss, DICE-RL w/o chunked RL, and ResFiT**. We find that DICE-RL outperforms both ablations, and both ablations outperform ResFiT. This supports the two core design choices directly. Chunked RL reduces the effective horizon, accelerates value backup and credit assignment, and encourages temporally coherent exploration rather than isolated micro-edits. BC regularization keeps exploration controlled and helps prevent collapse on long-horizon, precision-sensitive tasks.
>
> - EXPO: EXPO can underperform DICE-RL on long-horizon tasks because it lacks DICE-RL’s explicit BC regularization, making it more prone to critic error and drift from the prior; it updates a moving base policy online, whereas DICE-RL freezes the pretrained policy and learns only a lightweight residual; and DICE-RL uses chunk-level RL, which improves temporal consistency and long-horizon credit assignment.
>
> - DSRL: the reason DSRL is worse might be because RL does not optimize $a$ directly; it optimizes latent noise $w$, and reward is only realized after the noise->action map $a=\pi_{dp}(s,w)$. That makes credit assignment less direct and the optimization landscape potentially harder. Its two-stage critic, with $Q_W(s,w)\approx Q_A(s,\pi_{dp}(s,w))$, can also propagate value-estimation error. Finally, DSRL only does **steering** so it’s limited by the upper bound of the prior: it can improve BC, but only within behaviors reachable from the pretrained policy.
>
> - IBRL: IBRL may underperform DICE-RL because it only selects between a BC action and an RL action using the critic, rather than learning a local residual correction of the BC action itself. This makes fine, precise fixes harder. It is also a per-step RL method, whereas DICE-RL uses chunk-level residual RL, which is better suited to sparse, long-horizon manipulation. Finally, IBRL is built on a Gaussian BC policy, so it cannot leverage a rich generative BC prior as directly as DICE-RL.
>
> - DPPO: DPPO is substantially slower because it performs on-policy RL over a more extended horizon induced by the bi-level MDP formulation, which includes both the denoising chain and the task horizon. We found that if trained long enough, DPPO converges on simpler tasks, but its on-policy nature makes it less practical for real-robot finetuning, where data collection is expensive.
>
> Overall, DICE-RL combines several advantages that the baselines do not jointly have: **chunk-level RL** for stronger long-horizon credit assignment and temporally coherent exploration; **explicit BC regularization** for controlled exploration; **lightweight residual** finetuning on top of a frozen generative prior for stability and sample efficiency; and added mechanisms such as the BC-loss filter, multi-sample training, and best-of-K action selection that further improve finetuning.
>
> > I encourage the authors to ... large multi-task VLA policies …
>
> We thank the reviewer for the insightful question. We added (1) DICE-RL with a flow-matching policy on libero10 https://dice-rl-anonymous.github.io/#rebuttal-libero10 and (2) DICE-RL with π₀ on libero10 https://dice-rl-anonymous.github.io/#rebuttal-libero10-vla. DICE-RL improves the average succ. rate by 20% with only **~3000 online rollouts across all tasks, and is substantially more sample efficient than DSRL**. We also find that DICE-RL can further improve π₀ after supervised finetuning on libero10. For (2), we extract a 2048-dim latent feature from the raw obs and use it for finetuning.
> We believe DICE-RL is well suited for finetuning VLA policies: VLA pretraining provides broad semantic and behavioral coverage, while RL post-training sharpens the action distribution for precision, robustness, and long-horizon credit assignment. A promising direction for future work is to derive more compact VLA features that are predictive of task outcome (e.g., with mutual information), which could further improve efficiency. Another related future direction could be to identify an outcome-critical action subspace from online success/failure data, and learn the residual only there, instead of the full action.
>
> We thank the reviewer again for their time and effort. Please let us know if we can provide additional clarifications to improve our score. Your advice and feedback are highly appreciated!

---

> > ### Author Rebuttal · Reviewer_N9LR · 2026-04-03
> >
> > Thanks for the responses. Most of my concerns have been addressed properly.

---

> > > ### Author Response · Authors · 2026-04-04
> > >
> > > Thank you very much for your follow-up and for taking the time to reconsider the paper! We sincerely appreciate your careful reading and constructive feedback, which has been instrumental in improving the quality of our submission.

---

### Official Review · Reviewer_t2wZ · 2026-03-13

**Soundness:** 2
**Presentation:** 2
**Significance:** 3
**Originality:** 1
**Overall Recommendation:** 3
**Confidence:** 3

**Summary:**

This paper presents an approach for fine-tuning diffusion policies via “distribution contracting.” In particular, the paper freezes the base diffusion policy and instead learns a residual policy to correct actions from the base policy. The paper introduces a filtering mechanism to determine when to apply BC regularization on the residual policy’s updates. Empirically, the paper evaluates on both simulation, specifically Robomimic manipulation tasks, and real-robot experiments.

**Compliance With Llm Reviewing Policy:**

Affirmed.

**Final Justification:**

I maintained my score principally because of the lack of novelty and differentiation over the ResFit baseline. As the authors note in their rebuttal, the primary distinction from ResFit is adding chunking to the residual policy and adding a BC regularization loss. While the authors added ablations showing these additions improve over ResFit, both of these tools (action chunking and BC regularization losses) are standard, common practices, and I largely think of them as improvements to ResFit, rather than a standalone ICML paper. I appreciate the authors' rebuttal and they largely addressed my other concerns.

**Key Questions For Authors:**

1. Is the $K$ used for BoK action sampling during inference the same as the “K” used for multi-sample training? What is it in the ablation Figure 11?
2. Can you explain in more detail the difference between ResFit and DICE-RL?
3. Regarding the ResFit baseline, does the paper experiment with the RLPD-style offline/online mixing that you use for DICE-RL and DSRL?

**Limitations:**

yes

**Strengths And Weaknesses:**

Overall, my concerns are primarily with respect to soundness (specifically the implementation of the baselines) and originality (specifically the novelty over ResFit).
1. Regarding soundness, on line 264: “[when comparing to DSRL] DICE-RL can both optimize the latent noise and apply a learnable residual action correction…”. How does DICE-RL optimize the latent noise? From my understanding, DICE-RL does not modify the latent noised passed into the base policy; it only passes the latent noise into the residual policy.
2. Regarding the ResFit baseline, does the paper experiment with the RLPD-style offline/online mixing that you use for DICE-RL and DSRL?
3. The paper did not include results for DSRL without the offline/online mixing schedule. The original DSRL paper did not include the mixing schedule, so this deviation from the original implementation deserves justification (and specifically, a comparison of both variations).
4. The idea to measure metrics about the pretrained policy and data and observe their correlation with downstream performance is interesting. However, the actual metrics computed have issues. Regarding the measure of good mode coverage, this quantity is dependent on the learned value function, which undermines how valuable it is as a proxy metric for measuring the pretrained policy’s finetunability. Furthermore, if BC optimization is performed perfectly, then we would expect this measure to converge to 1. A more informative metric would be to estimate the pretrained policy’s Q function via MC rollouts in the environment, thus bypassing the critic dependence. While this requires environment interactions, these metrics are used for analysis, not for the actual algorithm.
5. The BoK action sampling procedure effectively results in another soft policy iteration - biasing more towards actions that optimize Q (and potentially further away from the behavior of the behavior/pretrained policy). A more informative ablation on BoK action sampling (Figure 11) would be to plot K on the X axis and success rate on the Y axis. Each line could be performance at a different number of environment steps. Thus, we can observe if there is a trend with K (e.g. using too large K reduces performance due to selecting over-optimistic actions).
6. As the performance of the pretrained policy improves, we’d expect to need less improvement over the base policy and thus fewer/smaller corrections. Methods like DSRL that simply modify the latent noise passed into the diffusion policy would likely become more effective. On the ablation over amount of pretraining data, including a baseline like DSRL would be informative.
7. The purpose of FILTER is to allow the RL component to improve over the base policy on actions that are genuinely than the base sample. I have two major concerns: the second component of FILTER seems likely to have high variance, since $\hat{G}$ is a Monte-Carlo estimate. Furthermore, FILTER seems redundant when using both $L_{RL}$ and $L_{BC}$. Early in training, it seems like FILTER will only “work” when a^pre is bad, since a^cur cannot outperform the MC return estimate in the replay buffer (if the critic is learned well). If the pretrained policy is trained well, we would expect $Q(s, a^{pre}) \approx \hat{G}(s)$. So the implementation of FILTER seems less about allowing for meaningful improvements over the base policy and more about allowing the residual policy to correct the pretrained policy when the pretrained policy does not appropriately learn the training data.
8. Pretraining is limited to single-task settings. In practice, the pretraining → post-training paradigm is typically used in multi-task settings. Extending the experiments to consider multi-task pretraining and single-task post-training seems like an important direction to consider.
9. On the originality side, from my understanding, the only difference between ResFit and DICE-RL algorithms seems to be the BC regularization.

---

> ### Author Rebuttal · Authors · 2026-03-30
>
> > Regarding soundness…
>
> We use “optimize” to mean best-of-K sampling. We will change it to “sample.”
>
> > Regarding the ResFit…RLPD-style…and DSRL?
>
> ResFiT already includes RLPD: Fig.5 of ResFiT shows removing offline data hurts. We added new ablations with several RLPD schedules, a fixed offline ratio of 0.1 and 0.0 https://dice-rl-anonymous.github.io/#rebuttal-rlpd. W/o offline data, DICE-RL generally shows only modest drops.
>
> > The paper did not…for DSRL without the offline…
>
> DSRL explicitly shows that offline data improves performance (Fig. 5), which they call offline-to-online adaptation. We first used DSRL w/o offline data and found slightly worse performance, so we add offline data as in DICE-RL.
>
> > The idea to measure metrics...
>
> We thank the reviewer. These metrics are critic-dependent, though empirically consistent across pretraining axes. We will clarify this in the revision.
>
> Their purpose is also broader than checkpoint comparison: they are meant to inform pretraining data collection. In particular, the analysis suggests that more consistent demos make DICE-RL finetuning easier. We also appreciate the suggestion to replace the critic with MC rollout estimates would be a stronger validation, and an important follow-up is to test whether the induced ranking is preserved when $Q_{\phi}$ is replaced by MC rollout returns.
>
> > The BoK action sampling…
>
> New ablation https://dice-rl-anonymous.github.io/#rebuttal-bok. For K=4,16,32,64, DICE-RL consistently outperforms w/o BoK. There is a slight drop at $K=64$, likely from overestimation, but multi-sample training keeps it robust to K.
>
> > As the performance of the pretrained policy improves …
>
> https://dice-rl-anonymous.github.io/#rebuttal-pretraining-data. In fact, the original DSRL paper uses the full pretraining dataset yet sample efficiency is limited even in state-space.
>
> > The purpose of FILTER...
>
> We appreciate the insightful question.
>
> On the variance of the second term, $\hat G_b(s)$ is not a MC estimate but the replay-attached MC return. It is noisy, but used only as a plausibility check for the value on the edited chunk. It enters the actor update only through the binary gate on the BC loss, not the Q-loss. We also use a negative margin, so the edited chunk stays below the replay return, making the gate conservative. We agree this is still heuristic and will clarify in revision.
>
> On redundancy with RL + BC loss, we respectfully disagree as the two losses can **conflict**: the RL term pushes away from the prior when the critic predicts improvement, while the BC term always pulls back. The filter makes this tradeoff selective & targeted, keeping the BC loss when the prior is desirable and removing it only when deviation is beneficial & reliable. This is especially important because $a^{\mathrm{pre}}$ comes from a stochastic generative prior, so samples at the same state can differ in quality.
>
> > Pretraining is limited to single task...
>
> We added **Libero10 results for DICE-RL with an FM policy** (https://dice-rl-anonymous.github.io/#rebuttal-libero10) and with **π₀** (https://dice-rl-anonymous.github.io/#rebuttal-libero10-vla). DICE-RL improves success by 20% with only ~3000 rollouts across tasks, and also improves π₀ after supervised finetuning on Libero10.
>
> > On the originality side ...
>
> DICE-RL differs from ResFiT in two key ways: it uses chunk-level residual policy and Q learning, whereas ResFiT uses a per-step residual actor-critic on top of a frozen chunked BC prior; and it adds an explicit BC loss to keep finetuning near the prior. DICE-RL also uses a BC-loss filter, best-of-K selection, and multi-sample training.
>
> Our new ablation https://dice-rl-anonymous.github.io/#rebuttal-resfit supports these choices: **DICE-RL outperforms DICE-RL w/o BC loss, DICE-RL w/o chunked RL, and ResFiT, while both ablations still beat ResFiT**. This shows chunked RL is already stronger than ResFiT’s per-step RL, and BC regularization provides an additional gain by preventing early prior collapse and stabilizing exploration.
>
> Although ResFiT emphasizes per-step reactivity, it also notes that chunking reduces the effective horizon and that n-step returns with n>1 matter on sparse, long-horizon tasks. For long-horizon, precise manipulation, chunked RL is preferable because it shortens the horizon, accelerates credit assignment, and explores coherent sequences rather than isolated micro-edits. This matches the large gains over ResFiT on Transport & Tool Hang.
>
> The BC loss keeps exploration controlled and within the prior’s support; w/o it, exploration is less constrained and can collapse.
>
> > Is the K used for BoK action...
>
> BoK and multi-sample training use different K’s, though they may be set equal. Fig.11 compares DICE-RL with and w/o BoK. We will clarify this in the revision.
>
> We thank the reviewer again for their time and effort. Please let us know if we can provide additional response to improve our score. Your advice and feedback are highly appreciated!

---

> > ### Author Rebuttal · Reviewer_t2wZ · 2026-04-03
> >
> > Thank you for your responses. I will maintain my score.

---

> > > ### Author Response · Authors · 2026-04-03
> > >
> > > Thank you again for your follow-up! We sincerely appreciate your detailed feedback, which has been instrumental in strengthening the submission. We are also grateful that you found the concerns fully resolved. In the rebuttal, we added targeted clarifications and new experiments addressing the main issues in the original review, including (1) the ResFiT novelty question and a detailed breakdown experiment, (2) the role of offline/online mixing, (3) DSRL with varying prertraining data, (4) the BoK ablation, results beyond single-task pretraining on (5) Libero10 and (6) π0, and the BC loss filter clarification. Since these were the core concerns behind the initial weak reject, we would be very grateful if you could reconsider the overall score to better reflect your updated assessment. If there are any remaining questions or concerns, we would be very happy to address them! Thank you again for your time and consideration!

---

### Official Review · Reviewer_24a2 · 2026-03-13

**Soundness:** 3
**Presentation:** 3
**Significance:** 2
**Originality:** 2
**Overall Recommendation:** 4
**Confidence:** 3

**Summary:**

DICE-RL treats RL as a post-training mechanism for a pretrained generative behavior policy rather than learning a robot policy from scratch. The method freezes the pretrained policy, learns a residual off-policy actor with selective BC-style regularization, and uses value-guided best-of-K action selection during interaction. The paper supports the approach with Robomimic experiments, a real-world belt assembly task, and appendix ablations on pretraining quality, convergence status, sampling, and implementation details.

**Compliance With Llm Reviewing Policy:**

Affirmed.

**Final Justification:**

My main concerns have been addressed with useful clarifications and additional empirical evidence.

1. **Method positioning and novelty.** The rebuttal clarifies that the contribution is not centered on the novelty of each individual component, but on a coherent post-training design guided by distribution contraction/sharpening. This framing makes the paper’s contribution clearer and more internally consistent.

2. **Theoretical claims.** The authors appropriately clarified that contraction is intended as a design rationale rather than a theorem-level guarantee. This substantially improves the rigor of the presentation and resolves my concern that the original wording was somewhat stronger than what is formally supported.

3. **Empirical support.** The added evidence, including the new real-world clarifications and the additional results beyond the original single-task setting, strengthens the empirical support of the paper and improves confidence in the practical usefulness of the method.

4. **Scope and presentation.** The rebuttal also addressed my concerns about overly strong wording and metric interpretation, including the title revision and the clarification of the scope of the finetunability-related metrics.

That said, my original reservations about the overall level of novelty and the lack of stronger theoretical guarantees are only partially alleviated, as these aspects are mainly clarified rather than fundamentally changed.

Overall, the rebuttal significantly improves the clarity, positioning, and empirical support of the work. Based on these improvements, I am inclined to slightly increase my assessment.

**Key Questions For Authors:**

- The filtered BC regularizer relies on a joint gating mechanism based on critic predictions and Monte Carlo return estimates. Why this heuristic is sufficient to reliably prevent exploitation of critic overestimation is still explained mainly at an empirical level rather than in a principled way.

- In the real-robot setting, the effects of human intervention frequency, early termination policies for failed trials, and the consistency of reward labeling are not analyzed in sufficient detail.

- The relationship between the proposed finetunability metrics—good coverage, bad coverage, and bad entropy—and final RL performance is intuitive, but the paper does not yet explain their threshold choices, cross-task robustness, or broader generalizability clearly enough.

**Limitations:**

Yes

**Strengths And Weaknesses:**

Strengths

- The paper is built around a well-defined problem setting, and its central claim is clearly articulated: in robotic scenarios with sparse rewards, long horizons, and limited online interaction budgets, RL is better suited as a post-training stage for pretrained generative behavior-cloning policies rather than as a mechanism for exploration from scratch.

- The method design is technically sound and shows strong engineering practicality. By freezing the pretrained generative policy, learning only a residual actor, and combining this with a TD3+BC-style objective, a filtered BC regularizer, and best-of-K value-guided selection, the method strikes a reasonable balance between preserving prior support and allowing necessary deviations.

- The experiments cover both state-based and pixel-based observations, multiple long-horizon Robomimic tasks, and one real-robot BeltAssembly task. In addition to final success rates, the main paper also reports training stability and sample efficiency, which makes the empirical evaluation more informative.

- The authors go beyond simply showing that the method works. They also systematically analyze which kinds of pretrained policies are more amenable to subsequent RL finetuning, including factors such as demonstration quality, checkpoint training stage, number of demonstrations, and metrics such as good mode coverage and bad mode entropy. This part provides additional insight into the method and adds interpretive value.

Weaknesses

- Most of the major components—residual policy correction, off-policy actor-critic learning, behavior regularization, value-guided candidate selection, and offline-online mixing—can be traced to existing lines of work. As a result, the paper reads more as a strong integrated solution for generative BC post-training than as a fundamentally new methodological contribution.

- The theoretical support is relatively limited. Although the paper emphasizes “distribution sharpening” and “contraction,” these notions function more as empirical characterizations and post hoc interpretations than as rigorously established theoretical guarantees. The explanations for stability and sample efficiency still rely primarily on experimental observations.

- If the phrase “skill mastering” in the title is intended as a strong claim, the current evidence remains somewhat limited. The main paper evaluates on four Robomimic tasks plus one real-world belt assembly task, and the task diversity, robot platforms, and reward settings are still fairly concentrated.

- The real-robot experiments are persuasive, but they rely on human supervision, manual resets, and binary success labels. The paper would benefit from a more explicit quantification of real-world wall-clock cost, the level of human involvement, and sensitivity to labeling consistency.

- The paper could also position itself more clearly with respect to recent work on generative policy post-training, including the limitations of fair comparison, as well as its relationship to broader robotics post-training and RLVR-style framings.

-The title of Section 5.5  “Ablations Studies” -> “Ablation Studies.”

---

> ### Author Rebuttal · Authors · 2026-03-30
>
> > Most of the major ...
>
> We appreciate the reviewer’s perspective. Our claim is not each ingredient is new in isolation, but distribution sharpening and trajectory contraction provide **a guiding principle for post-training**, motivating best-of-K action selection, the BC-loss filter, and multi-sample training. Rather than a loose aggregation of prior ideas, we see the contribution as a coherent RL post-training design for generative BC policies. Empirically, this design enables new capabilities on challenging long-horizon manipulation, e.g., solving Tool Hang from image obs with only 50 demos and >90% success.
>
> > The theoretical support is ...
>
> We agree that our paper does not prove theorem-level guarantees for sharpening or contraction. Instead, these observations provide the design principles behind DICE-RL: sharpening high-value modes and making closed-loop behavior more contractive motivates multi-sample expectation training, best-of-K selection, and the BC-loss filter, whose benefits are validated in ablations. We view contraction as a theoretical rationale, not a formal guarantee. For closed-loop dynamics
>
> $$
> s_{t+1} = f(s_t, \pi(s_t))
> $$
> . if finetuning reduces the local sensitivity
> $$
> \\left\\| \\frac{\\partial f}{\\partial s} + \\frac{\\partial f}{\\partial a}\\frac{\\partial \\pi}{\\partial s} \\right\\|
> $$
>
> in failure-prone regions, nearby trajectories should converge faster and disturbances should attenuate rather than amplify. Our trajectory-divergence and perturbation analyses test this local-stability hypothesis.
>
> > The phrase “skill mastering”...
>
> Thank you for the suggestion! We will change it to “skill refinement.”
>
> > The real-robot ...
>
> We appreciate the feedback. In the revision, we will clarify **wall-clock cost, label sensitivity, and human involvement** in the real-world experiments. Performance is most sensitive to success-label consistency, since all failures receive reward 0; automatic reward detection can help reduce this variance. We have added **a new light bulb insertion task with automatic reward detection** https://dice-rl-anonymous.github.io/#rebuttal-real-world-reward and a timelapse video for a gear insertion task https://dice-rl-anonymous.github.io/#rebuttal-real-world-timelapse. More details about wall-clock time and reward labeling are provided at the links.
>
> > The paper could also position...
>
> We appreciate this suggestion. We will clarify DICE-RL relative to prior post-training methods: what is optimized (a stochastic generative prior), how improvement is constrained (explicit tethering to the pretrained prior), and which regime it targets (long-horizon, precise manipulation under sparse reward). We will relate DICE-RL to broader robotics post-training and RLVR-style framings. We added results for **DICE-RL with a VLA on Libero10**, connecting our method to broader robotics post-training:https://dice-rl-anonymous.github.io/#rebuttal-libero10-vla.
>
> > The filtered BC...
>
> We acknowledge that the filter is not a theorem-level guarantee against Q overestimation, but an empirically motivated safeguard: it keeps BC regularization unless there is evidence that deviating from the pretrained chunk is both beneficial and reliable. We acknowledge that this is supported mainly empirically rather than formally. A more principled direction as future work could be to replace the heuristic gate with a certificate-based safe-switching rule based on a prior-conditioned recoverability funnel, relaxing the BC penalty only when the edited chunk is certified to stay within the pretrained policy’s recoverable tube throughout the chunk and to improve a local contraction/recoverability certificate relative to the pretrained chunk. This would place the filter on firmer theoretical footing as a principled criterion for when leaving the prior is justified.
>
> > The relationship between …
>
> We appreciate the reviewer’s comment and will clarify the threshold choice, robustness, and scope of these metrics. The threshold is fixed across all tasks ($\alpha = 0.5$), not tuned per task. We use
> $$
> Q_\phi(s,\pi_{\mathrm{pre}}(s,z)) \ge \alpha \hat G_b(s)
> $$
> so that the notion of a “good” sample is normalized by the anchor-state return and is comparable across tasks with different horizons. The metrics track downstream finetuning behavior consistently across multiple pretraining axes. We will be more explicit about generalizability: because these metrics rely on a fixed critic-based evaluator, they should be understood as analysis under a fixed downstream RL setup, rather than universal, critic-independent metrics of finetunability. We will make this scope clearer, test threshold sensitivity, and verify if the rank ordering is preserved under alternative metrics, such as MC return from pretrained policy rollouts.
>
> We thank the reviewer again for their time and effort. Please let us know if we can provide additional clarifications to improve our score. Your advice and feedback are highly appreciated!

---

> > ### Author Rebuttal · Reviewer_24a2 · 2026-04-04
> >
> > I thank the authors for the detailed and constructive rebuttal. My main concerns have been addressed with useful clarifications and additional empirical evidence.
> >
> > 1. **Method positioning and novelty.** The rebuttal clarifies that the contribution is not centered on the novelty of each individual component, but on a coherent post-training design guided by distribution contraction/sharpening. This framing makes the paper’s contribution clearer and more internally consistent.
> >
> > 2. **Theoretical claims.** The authors appropriately clarified that contraction is intended as a design rationale rather than a theorem-level guarantee. This substantially improves the rigor of the presentation and resolves my concern that the original wording was somewhat stronger than what is formally supported.
> >
> > 3. **Empirical support.** The added evidence, including the new real-world clarifications and the additional results beyond the original single-task setting, strengthens the empirical support of the paper and improves confidence in the practical usefulness of the method.
> >
> > 4. **Scope and presentation.** The rebuttal also addressed my concerns about overly strong wording and metric interpretation, including the title revision and the clarification of the scope of the finetunability-related metrics.
> >
> > That said, my original reservations about the overall level of novelty and the lack of stronger theoretical guarantees are only partially alleviated, as these aspects are mainly clarified rather than fundamentally changed.
> >
> > Overall, the rebuttal significantly improves the clarity, positioning, and empirical support of the work. Based on these improvements, I am inclined to slightly increase my assessment.

---

> > > ### Author Response · Authors · 2026-04-04
> > >
> > > Thank you very much for your thoughtful follow-up and for taking the time to reconsider the paper! We sincerely appreciate your careful reading, constructive feedback, and updated assessment. We are grateful for your time and detailed suggestions on how to further improve the work.

---

### Official Review · Reviewer_b6jW · 2026-03-15

**Soundness:** 2
**Presentation:** 3
**Significance:** 3
**Originality:** 2
**Overall Recommendation:** 3
**Confidence:** 3

**Summary:**

This paper proposes DICE-RL, a distribution contractive reinforcement learning (RL) finetuning framework for pretrained generative robot behavior cloning (BC) policies. It freezes the pretrained BC backbone, learns a lightweight action-chunk-level residual policy, and integrates a selective BC-loss filter, multi-sample training and value-guided action selection for stable policy refinement. Experiments on Robomimic benchmarks and real robot assembly tasks show DICE-RL outperforms mainstream baselines with higher success rate, stability and sample efficiency.

**Compliance With Llm Reviewing Policy:**

Affirmed.

**Final Justification:**

Although the author has addressed some of the issues I raised, others cannot be resolved in the short term, so I have decided to keep the current “weak reject” score unchanged.

**Key Questions For Authors:**

Questions:
1. Could the authors explain the rationale for adopting the per-chunk residual design instead of the widely used per-step residual scheme in related works like ResFit?
2. The paper notes that ResFit does not impose an explicit BC-style regularization during finetuning as the key difference from DICE-RL. Could the authors explain the core reasons for the performance gap between DICE-RL and ResFit under the setting of β=0 (i.e., without BC filter loss)?
3. Could the authors discuss the potential adaptability of the DICE-RL framework to finetuning large multi-task VLA models, and what key modifications may be needed for this application

**Limitations:**

The authors should include a discussion of limitations in the paper.

**Strengths And Weaknesses:**

Strength:
1. The framework strategically integrates the strengths of prior residual correction, distillation-style regularization and steering-style value-guided selection, with a well-motivated "distribution contraction" design highly practical for real-world robotic deployment.
2. The evaluation is comprehensive, covering simulation and real robot tasks across state and pixel inputs, with systematic ablations validating core module effectiveness and ensuring result reproducibility.
3. The work reveals the sharpening-contraction effect of RL finetuning on pretrained action distributions, and provides actionable guidance for BC policy pretraining and downstream finetuning.

Weakness:
1. The rationale for the core design choice of per-chunk residual is not sufficiently justified. The highly relevant baseline ResFit adopts a per-step residual design and qualitatively discusses the advantages of per-step over per-chunk residual in its work, while this paper does not provide an explicit explanation for choosing the per-chunk scheme.
2. The innovation boundary between the proposed method and the most relevant prior work ResFit needs further clarification. The current ablation study does not fully disentangle the performance impact of the core BC-loss filter from other design differences, and it is unclear how the method would perform against ResFit when the core module is removed.
3. The method is only validated on single-task BC pretraining and finetuning settings, and its applicability to multi-task VLA model finetuning (as explored in CO-RFT, Huang et al., 2025) has not been discussed, leaving the generalizability and scalability of the framework to be further explored.

---

> ### Author Rebuttal · Authors · 2026-03-30
>
> > (1) The innovation boundary…ResFit when the core module is removed. (2) The rationale for the core design choice of per-chunk, choosing the per-chunk scheme. (3) The paper notes that ResFit …β=0 (i.e., without BC filter loss)
>
> We thank the reviewer for these insightful questions. We organize our response into four points.
> - On the innovation boundary w.r.t. ResFiT. The core difference is **two-fold**: (i) DICE-RL performs **chunked residual action learning with chunked value learning**, whereas ResFiT uses a per-step residual actor and critic; (ii) DICE-RL adds an **explicit BC regularization loss** to keep finetuning close to the prior. In addition, DICE-RL also includes design choices that further improve performance, such as BC-loss filter, best-of-K action selection, and multi-sample training. ResFiT itself does not benchmark against chunked residual actor + critic, so its paper itself does not establish per-step RL is superior to chunk-level RL.
>
> - New experiment. Our new ablation at https://dice-rl-anonymous.github.io/#rebuttal-resfit cleanly justifies the two core differences. We compare DICE-RL, DICE-RL w/o BC loss, DICE-RL w/o chunked RL, and ResFiT. **DICE-RL is better than both DICE-RL w/o BC loss and DICE-RL w/o chunked RL, which are better than ResFiT**. This shows 1) even w/o BC regularization, the chunked RL design is already stronger than ResFiT with per-step actor + critic design, so chunked residual action learning and chunked value learning are a major source of the gain; 2) BC regularization provides an additional, complementary benefit by preventing early collapse of the prior and keeping exploration stable. Although the pretrained BC policy trained using the ResFiT codebase starts from a higher initial performance, it quickly collapses during the early stage of finetuning. In short, the gap to ResFiT comes from both ingredients: chunk-level RL for better temporal credit assignment and coherent exploration, and BC regularization for controllability and stability. Below we provide a detailed explanation of why the two differences make DICE-RL better.
>
> - Why chunked RL can be better for our specific problem setting. ResFiT argues that per-step residuals provide more reactive, fine-grained correction, which is a reasonable design choice. However, our results on long-horizon, precise tasks show that chunked RL can in fact **provide exactly the desired precision and stability**. The reasons are 1) chunked value learning reduces the effective horizon, speeds up value backup and credit assignment [1], and 2) chunked residual learning (together with bc loss) explores through temporally coherent action chunks, making exploration more efficient and effective and mitigating compounding error [2]. This is in fact consistent with ResFiT’s own observations: it explicitly states that action chunking alleviates compounding error by reducing the task horizon, reports that on its sparse-reward task approaches without action chunking fail, and finds that n-step returns>1 are important. Together, these points suggest that the key issue is not only reactivity, but also how efficiently value propagates through long-horizon sparse-reward tasks.
>
> - On the role of BC regularization. The BC loss keeps exploration controllable and within the support of the prior. When it is removed, exploration becomes much less constrained and can collapse, especially in sparse-reward continuous control. This is precisely what we observe for ResFiT on the longer-horizon tasks: lacking explicit BC regularization, it can drift away from the pretrained prior very early.
>
> > “The method is only validated on single-task … framework to be further explored.”
>
> We thank the reviewer for this insightful comment. We added (1) DICE-RL with a flow-matching policy on libero10 (https://dice-rl-anonymous.github.io/#rebuttal-libero10) and (2) DICE-RL with π₀ on libero10 (https://dice-rl-anonymous.github.io/#rebuttal-libero10-vla). The results show that DICE-RL improves the average succ. rate by 20% with only **~3000 online rollouts across all tasks, and is substantially more sample efficient than DSRL**. We also find that DICE-RL can further improve π₀ after supervised finetuning on libero10. For (2), we extract a 2048-dim latent feature from the raw obs and use it for finetuning. A promising direction for future work is to derive more compact VLA features that are predictive of task outcome (e.g., with mutual information), which could further improve the efficiency of DICE-RL.
>
> We thank the reviewer again for their time and effort. Please let us know if we can provide additional clarifications to improve our score. Your advice and feedback are highly appreciated!
>
> [1]. Li, Q., Zhou, Z., & Levine, S. (2025). Reinforcement learning with action chunking. arXiv preprint arXiv:2507.07969.
>
> [2] Simchowitz, M., Pfrommer, D., & Jadbabaie, A. (2025). The pitfalls of imitation learning when actions are continuous. arXiv preprint arXiv:2503.09722.

---

> > ### Author Rebuttal · Reviewer_b6jW · 2026-03-31
> >
> > I thank the Authors for their response. I will keep my score unchanged.

---

> > > ### Author Response · Authors · 2026-03-31
> > >
> > > Thank you again for your thoughtful follow-up! We really appreciate that you found the concerns **fully resolved**. Your feedback has been instrumental in strengthening our submission. In the rebuttal, we added new experiments and clarifications directly addressing the main points behind the initial weak reject: a clean comparison against ResFiT disentangling chunked RL and BC regularization, a clearer justification for the per-chunk design, and new results showing applicability beyond the single-task setting on Libero10 / π₀​. Since these were the core concerns in the original review, we would be very grateful if you could reconsider the overall score to reflect your updated assessment. Thank you again for your time and consideration!
> > >
> > > Thank you,
> > > Authors

---

### Decision · Program_Chairs · 2026-04-30

**Decision:**

Accept (regular)

**Comment:**

### Summary
This paper presents a practical recipe for fine-tuning a behavioral cloning (BC) robotics policy using reinforcement learning (RL). Empirical evaluation is performed on some robomimic benchmarks as well as a real robot task (belt assembly).

### Meta-Review
The proposed recipe for fine-tuning the robot control policy after pre-training via BC seems to be empirically effective, and evaluation on a real robot task lends credence to the proposed approach. Ablations and experimental probes lay bare how the design choices affect the performance of the proposed approach. There should be more discussion on the comparison of the proposed technique with ResFit, since not seeing the difference in the two techniques was a point that the reviewers brought up.